



# Tracing changes in atmospheric moisture supply to the drying Southwest China

Chi Zhang[1], Qiuhong Tang[1,5], Deliang Chen[2], Laifang Li[3], Xingcai Liu[1], Huijuan Cui[4]

[1]Key Laboratory of Water Cycle and Related Land Surface Processes, Institute of Geographic Sciences and Natural Resources Research, Chinese Academy of Sciences, Beijing, China

[2]Regional Climate Group, Department of Earth Sciences, University of Gothenburg, Gothenburg, Sweden

[3]Earth and Ocean Sciences, Nicholas School of the Environment and Earth Sciences, Duke University, Durham, USA

[4]Key Laboratory of Land Surface Pattern and Simulation, Institute of Geographic Sciences and Natural Resources Research, Chinese Academy of Sciences, Beijing, China

[5]University of Chinese Academy of Sciences, Beijing, China

*Correspondence to*: Qiuhong Tang (tangqh@igsnrr.ac.cn)

**Abstract.** Precipitation over Southwest China (SWC) has significantly decreased during 1979–2013. The summer months from July to September contributed the most to the decrease of precipitation. By tracing moisture sources of summer precipitation over the SWC region, it is found that most moisture originates in the monsoon region. The major moisture contributing area is divided into an extended west region, SWC, and an extended east region. The extended west region is mainly influenced by the South Asian Summer Monsoon (SASM) and the westerlies, while the extended east region is mainly influenced by the East Asian Summer Monsoon (EASM). The extended west, SWC, and extended east regions contribute 48.2%, 15.5%, and 24.5% of moisture for the SWC precipitation, respectively. Moisture supply from the extended west region decreased at a rate of -23.6 mm decade$^{-1}$ whereas that from the extended east increased at a rate of 4.2 mm decade$^{-1}$, resulting in an overall decrease of moisture supply. Further analysis reveals that the decline of summer precipitation is mainly caused by change in the stationary component rather than the transient component of the moisture transport over the SWC region. The dynamic process (i.e. change in circulation) rather than the thermodynamic process (i.e. specific humidity) is dominant in affecting the stationary moisture transport. A prevailing easterly anomaly of moisture transport that weakened moisture supply from the Indian Ocean is to a large extent responsible for the precipitation decrease over the SWC region.

## 1 Introduction

Frequent and severe droughts hit Southwest China (SWC) over the last decades with record-breaking events in the summer of 2006 and 2011, which had caused great losses to the society. The intensified drought is characterized by the persistent deficit of precipitation (Wang et al., 2015b), and has attracted much attention (e.g. Barriopedro et al., 2012; Feng et al., 2014; Wang et al., 2015b; Tan et al., 2016; He et al., 2016; Zhang X. et al., 2017).





Many studies have analyzed the meteorological conditions that caused the extremely low precipitation for individual drought cases (e.g. Li et al., 2011; Lu et al., 2011; Yang et al., 2012). Taking the drought of summer 2006 as an example, a stronger Western Pacific Subtropical High (WPSH) was found to lie anomalously northward and westward (Li et al., 2011). Under the direct control of WPSH, descent flows prevailed over SWC and the moisture transport from the Bay of Bengal

(BOB) and South China Sea (SCS) was suppressed (Liu et al., 2009; Li et al., 2011). Further analysis revealed that the active convection over the Philippines and the weaker-than-normal heat source of the Tibetan Plateau drove the strengthened WPSH to shift northward and westward (Li et al., 2011). Meanwhile, a weak blocking high in the Ural Mountains and a shallow East Asian trough facilitated a stronger-than-normal zonal circulation in the mid-latitudes, which hindered the intrusion of cold air into SWC (Zou and Gao, 2007). In summary, the configuration of the large-scale subtropical and mid-

latitude circulations was unfavorable for the warm-moist air from the south and cold-dry air from the north to converge over SWC, and thus produced the severe drought.

Some recent studies have endeavored to investigate the mechanisms causing the SWC drying in a long climatological perspective. Using stalagmite record as a proxy, Tan et al. (2016) found the period of 2009-2012 was the driest ever since 1760 AD in SWC. They further attributed the drying trend to the warming of Tropical Ocean, which had reduced the land-

15 sea thermal gradient and the amount of moisture transported from the BOB. In another study, the possible influence of sea surface temperature (SST) in tropical Northwest Pacific (NWP) on the autumn precipitation in SWC was investigated (Wang et al., 2015a). It was found that the warm SST in NWP had likely contributed to the dry condition in SWC in recent decades.

Although previous studies have deepened our understanding of the SWC drying through attributing individual/general drought events or long-term precipitation trend to some probable causes, few of them have analyzed the changes in the

20 precipitation moisture sources of this region. Tracing the precipitation moisture sources can reveal the origin of moisture for precipitation (Gustafsson et al., 2010; Zhang C. et al., 2017), thus providing insights on how the changes in moisture sources may affect precipitation in SWC. This study intends to identify changes in moisture sources of the SWC precipitation during the last several decades and to investigate the possible mechanism of the SWC drying.

## 2 Data and Methodology

### 2.1 Data

The reanalysis of the European Centre for Medium-Range Weather Forecasts (ECMWF) Interim (ERA-I hereafter) was used to calculate precipitable water and moisture flux (Dee et al., 2011; Trenberth, 1991). The ERA-I data have a spatial resolution of 1.5° × 1.5° grid cell, and the time span of the ERA-I in this study is from 1979 to 2013. The data include the 6-h wind and moisture content at multiple levels from 200–1000 hPa, and surface pressure.

Due to the existing limitation with precipitation and evaporation estimates in reanalysis products, the ground-based 0.5°-gridded daily precipitation dataset from the China Meteorological Administration (CMA) was used (Zhao et al., 2014; Zhao and Zhu, 2015). The 3-h 1°-gridded evaporation fields from the Community Land Model in the Global Land Data





Assimilation Systems (GLDAS, Rodell et al., 2004) dataset were used. Over the ocean, the evaporation fields in ERA-I reanalysis were used directly, since there is no alternative estimate.

## 2.2 Water Accounting Model

The Water Accounting Model (WAM) is an Eulerian model on moisture recycling, which can quantify the moisture source-sink relations between evaporation and precipitation by tracking moisture forward or backward in time (van der Ent et al., 2010; van der Ent and Savenije, 2011). In this study, moisture backtracking was applied to track the moisture origins of and their changes with the SWC precipitation. The algorithm is briefly described as follows.

The input of WAM includes precipitation, evaporation, and atmospheric data (precipitable water and moisture transport). The fallen precipitation in the target area was assumed to return to the air as "tagged water" in the model. The tagged water was mixed into the precipitable water with a ratio of $r$, which means only $r$ proportion of the precipitable water would finally fall into the target area. When it reverses the way back along the transport path, a certain amount of moisture, which is evaporated from the sources in the path, would fall into the target area. The ratio of that certain amount is also $r$. Taking the first source grid for example, it evaporates an amount of $e$ into the air at this time step. At the same time, the mixed ratio is $r$, and then the only $e*r$ would finally fall into the target area. The direct contribution from the grid at this time step is $e*r$. The tagged water would reduce the same amount of $e*r$ and move on to the next source grids till all the tagged water is depleted. By then, the total moisture contribution from each grid can be summed to produce a spatial distribution of moisture contributed to the precipitation in the target area.

The time step of WAM was set to 0.5 h for 1.5° grid in this study as in van der Ent et al. (2010) and van der Ent and Savenije (2011). The 6-h atmospheric data of precipitable water and moisture flux were linearly interpolated into the 0.5-h time step. For evaporation, the 1°-gridded GLDAS data were first interpolated into the 1.5° grid over the land, and then were merged with the ocean evaporation from the ERA-I reanalysis. The merged evaporation, which was 3-h accumulated, was divided equally into the 0.5-h time step. For precipitation, in order to reflect the diurnal cycle, the daily CMA and 3-h ERA-I precipitation fields were merged. The CMA precipitation was firstly transformed to the same spatial resolution as ERA-I by taking the means of the 0.5° grids that fell into the 1.5° grid. Then both monthly CMA and ERA-I precipitations were calculated. By setting the monthly CMA precipitation as a norm, all ERA-I precipitation during a month was rescaled proportionally. Finally, the rescaled 3-h-accumulated ERA-I precipitation was equally distributed over the 0.5-h time step.

When the WAM was applied at a monthly scale, a large amount of tagged water may be left in the air after one month's tracking rather than allocated to the surface sources. In this case, the backtracking would continue to run another 30 days with no input of precipitation to ensure that 95% of the monthly precipitation moisture returned to the surface.

## 2.3 Decomposing moisture transport

To further understand the change of moisture transport in association with the change of moisture origin, the monthly vertically integrated moisture flux was decomposed into a stationary component and a transient component (Li et al., 2013a).





The stationary component is the monthly mean moisture transported by the monthly mean flow, while the transient component is the transient moisture transported by the transient eddies. The stationary component was calculated using the average of 6-h values in each month. The transient component was calculated using the 6-h deviations from the monthly mean. The fluctuation of the stationary component can be further expressed by a thermodynamic and a dynamic term
(Seager et al., 2010; Li et al., 2013a). The thermodynamic (dynamic) component was determined by the changes in specific humidity (wind velocity) and thus represents the thermodynamic (dynamic) contribution (Li et al., 2013a). The variations in stationary and transient components and thermodynamic and dynamic terms were analyzed.

## 3 Results and discussion

Figure 1a shows the annual precipitation trends from 1979–2013 calculated from the CMA gridded precipitation over the
10 SWC region as marked out by the red box. The SWC precipitation has experienced a declining trend in recent decades. The area-averaged annual SWC precipitation has decreased significantly with a rate of -2.72 mm yr$^{-1}$ (Fig. 1b). Table 1 provides the monthly precipitation trends during 1979–2013. It shows that the monthly trends from March to May are positive, although they are not statistically significant. The decreasing trends are the largest in summer with rates of -0.5, -1.1, and -1.0 mm yr$^{-1}$ in July, August, and September, respectively. The decreasing trend is statistically significant at the 6% (1%)
level in August (September). The total summer precipitation decreased significantly at the 5% level with a rate of -2.57 mm yr$^{-1}$ (Fig. 1c), which accounts for 94.5% of the annual trend. Since the summer precipitation change accounts for a major share of annual precipitation trend, the analysis below will focus on the summer months.

### 3.1 Moisture origin and the trend in moisture contribution

The climatological moisture contributions from the source grids in summer and their trends during 1979–2013 are shown in
Fig. 2. The major moisture contributing region, i.e., grids with contribution over 0.8 mm yr$^{-1}$, are marked out (Fig. 2a), where 88.3% of summer precipitation moisture in SWC comes from. As shown in Fig. 2, the farther away from the target region, the lower intensity of moisture is contributed to the target (Zhang C. et al., 2017). The moisture contribution intensity lapses slowly to the southwest and southeast, indicating that the monsoon regions provide considerable precipitation moisture to SWC. In contrast, the intensity lapses rapidly to the north, suggesting that little precipitation moisture originates from the
north. To the west of the SWC region, the intensity of moisture contribution is low in the dry area such as the Middle East but relatively high in surrounding wet areas such as the Caspian Sea and the Red Sea, suggesting that humid region provides more moisture than arid region.

As the moisture contribution trends show an opposing pattern in the west and east (Fig. 2b), the major moisture contributing region is divided into three regions, namely the extended west, SWC, and the extended east regions. The
30 extended west region covers an area west and southwest to SWC, and the extended east region covers an area east to SWC and a part of the Indian Ocean. Figure S1 shows the climatological moisture transport from July to September. It indicates





moisture from the extended west region largely enters the western and southern borders of SWC whereas moisture from the extended east region enters the eastern border via a route through the SCS. Moisture from the extended west region is likely affected by the South Asian Summer Monsoon (SASM) and the westerlies, while that from the extended east region is likely affected by the East Asian Summer Monsoon (EASM). When summed over regions, the extended west, SWC, and the

extended east regions contribute 48.2%, 15.5%, and 24.5% of the total precipitation moisture, respectively. The moisture contribution directly from the Tibetan Plateau is only 11.5%, much less than that from the SASM and EASM regions (Yao et al., 2012; Huang and Cui, 2015).

Figure 2b shows the trend of moisture contributed to the SWC summer precipitation from different regions during 1979–2013. Moisture supply from most of the extended west region experienced a decreasing trend of -23.6 mm decade$^{-1}$,

accounting for 91.7% of the SWC precipitation trend, while that from most of the extended east region experienced an increasing trend of 4.2 mm decade$^{-1}$. The trend of moisture contributed from SWC is -1.2 mm decade$^{-1}$, accounting for 4.6% of the SWC precipitation trend. It suggests that change in local recycling played a minor role in the precipitation decrease over SWC.

Figure 3 shows the changes in moisture contribution and moisture transport in July, August, and September between the

first and last ten years of the period of 1979–2013. Overall, there is an apparent decline of moisture supply from the west and southwest regions to SWC in all the three months. The area with the largest decline of moisture contribution includes the Indian subcontinent and Indochina over the land and the BOB over the sea. Compared with the moisture transport in first ten years, more moisture from the Indian Ocean has been routed to the northern Indian subcontinent or the Tibetan Plateau, rather than into SWC in the last ten years. Consequently, moisture contribution influenced by the SASM is weakening. In

contrast, moisture contribution has increased in many parts of the extended east region. In July, the area with increased moisture contribution includes the northern Central Indian Ocean, SCS, and a northeastern area of SWC. It looks like that more moisture from the northern Central Indian Ocean has been routed to SWC via SCS in the last decade of 1979–2013. In August and September, the main area with increased moisture contribution is located to the east and south of SWC, while a part of the northern Central Indian Ocean also contributed more moisture to SWC compared to the first ten years. The

prevailing easterly moisture transport in the recent decade in South China support an enhanced contribution of the SWC precipitation moisture from the EASM region. The southern part of SWC that is largely affected by the SASM and westerlies experienced a decrease of moisture contribution.

### 3.2 Thermodynamic and dynamic control of moisture transport

The role of moisture transport is further investigated by analyzing the relations between moisture divergence and

precipitation over SWC during 1979–2013. Correlation coefficients between moisture divergence and precipitation over SWC are calculated, which turn out as -0.87, -0.88, and -0.74 for July, August, and September, respectively. The close correlation indicates that remote moisture transport is more important than local surface evaporation in regulating the interannual variation of summer precipitation (Li et al., 2013b).





Figure 4 shows the 35-year climatology and time series of moisture divergence over SWC and their stationary and transient parts. The moisture divergences over SWC are negative as expected (see also Fig. S1). The stationary component of moisture divergence is also negative. The transient component is, however, positive. It indicates the mean flow converges moisture into SWC, while the transient eddies diverge moisture. The magnitude of the transient component is about 30% of

that of the stationary component, further suggesting that the change in the stationary component plays a major role in changing the moisture divergence over SWC (Fig. 4a). During 1979–2013, the stationary component increased, and the transient component decreased, resulting in an increasing trend of the moisture divergence (Fig. 4b). It suggests that the change in the mean flow rather than the transient eddies has led to the decrease of SWC summer precipitation.

Figure 5 shows the changes in thermodynamic and dynamic components of the stationary moisture transport in summer

over SWC during 1979–2013. The variation of the thermodynamic component is small compared with that of the dynamic component, suggesting that the dynamic process, i.e., change in atmospheric circulation (wind), exerted a dominant influence on the variation of moisture transport. The dynamic component shows an obvious increasing trend during 1979-2013, while the thermodynamic component shows a small negative trend. The increase of moisture divergence, i.e. decrease of moisture convergence, by the dynamic component is in line with the decreasing precipitation (Table 1). The correlation

coefficients between the dynamic component of moisture transport and precipitation over SWC are calculated. Based on the correlation coefficients, the dynamic component explains 80%, 81%, and 58% of the precipitation variances for July, August, and September, respectively. It confirms the dominant role of the dynamic process in regulating the precipitation change in SWC. The interannual variation in the SASM net precipitation is also dominated by the dynamic process (Walker et al., 2015). This suggests that the dominant role of the dynamic process in moisture transport and regional precipitation prevails

over a very large area.

Figure 6 compares the dynamic component in summer (July, August, and September) between the first and last ten years of the period of 1979–2013. There is an overall positive anomaly of moisture divergence over SWC with an easterly anomaly of moisture transport. Though there is a southwesterly anomaly of moisture transport from the Indian Ocean to the SWC direction in July and September, it does not contribute moisture transport to SWC because the anomaly ends on the

south of the Tibetan Plateau. There is an easterly anomaly along the southern edge of the Tibetan Plateau, routing the moisture transport to the northern Indian subcontinent instead of the SWC region. The anomaly of moisture divergence, dynamically caused by the changes in circulation, is generally negative in the Indian subcontinent but positive in SWC (Tan et al., 2016). The prevailing easterly anomaly of moisture transport and pronounced regional anomalies of moisture divergence over SWC are likely to result from the change in the Asian summer monsoon system (Wei et al., 2014), which

might be related to recent Pacific cooling and Indian Ocean warming (Ueda et al., 2015).



**4 Conclusions**

Summer precipitation over SWC has decreased during 1979–2013. By tracing moisture origin of precipitation in the summer months (July, August, and September) and by analyzing the variations of moisture transport to SWC, we came to the following conclusions.

(1) Most moisture for the SWC summer precipitation originates from the Asian monsoon regions. The westerlies play a secondary role in supplying moisture. The extended west region, SWC, and the extended east region contributes 48.2%, 15.5% and 24.5% of moisture to SWC summer precipitation, respectively. The Tibetan Plateau region contributes a small portion (11.5%) of the precipitation moisture.

   (2) The decrease in the summer precipitation is mainly attributed to the reduced moisture supply from the extended
west region. Moisture supply from the extended west region has decreased at a high rate (-23.6 mm decade$^{-1}$), and that from the extended east has increased at a low rate (4.2 mm decade$^{-1}$), resulting in an overall decrease of the moisture supply.

   (3) The change in the stationary component has reduced moisture transport into SWC whereas the change in transient component has increased moisture transport in summer during 1979–2013. The dynamic process (i.e., change in circulation) is more important than the thermodynamic process (i.e. specific humidity) in affecting the precipitation. A prevailing
easterly anomaly of moisture transport that weakened moisture supply from the Indian Ocean is mainly responsible for the decrease of the SWC precipitation. The change in circulation is likely related to the recent sea surface temperature change and need further investigation.

**5 Data availability**

The ERA-I data are available from the European Centre for Medium-Range Weather Forecasts (ECMWF). The GLDAS data
are available from the NASA Goddard Earth Sciences Data and Information Services Center (GES DISC).

*Competing interests*. The authors declare that they have no conflict of interest.

*Acknowledgements*. This work was supported through the National Natural Science Foundation of China (41425002), the
25 Key Research Program of the Chinese Academy of Sciences (ZDRW-ZS-2016-6-4), and the National Youth Top-notch Talent Support Program in China. Support from Swedish VR, STINT, BECC, MERGE and SNIC through S-CMIP are also acknowledged.



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





**Table 1. The monthly precipitation trends (mm decade$^{-1}$) in SWC during 1979 to 2013. The P-values of the trends were calculated based on the two-tailed Student's t-test.**

| Month | Jan | Feb | Mar | Apr | May | Jun | Jul | Aug | Sep | Oct | Nov | Dec |
|---|---|---|---|---|---|---|---|---|---|---|---|---|
| Trend | 0.2 | -1.4 | 1.3 | 0.1 | 3.7 | -1.9 | -5.0 | -10.7 | -10.0 | -1.6 | -1.4 | -0.5 |
| P-value | 0.79 | 0.27 | 0.41 | 0.97 | 0.28 | 0.50 | 0.32 | 0.06 | 0.01 | 0.55 | 0.49 | 0.61 |



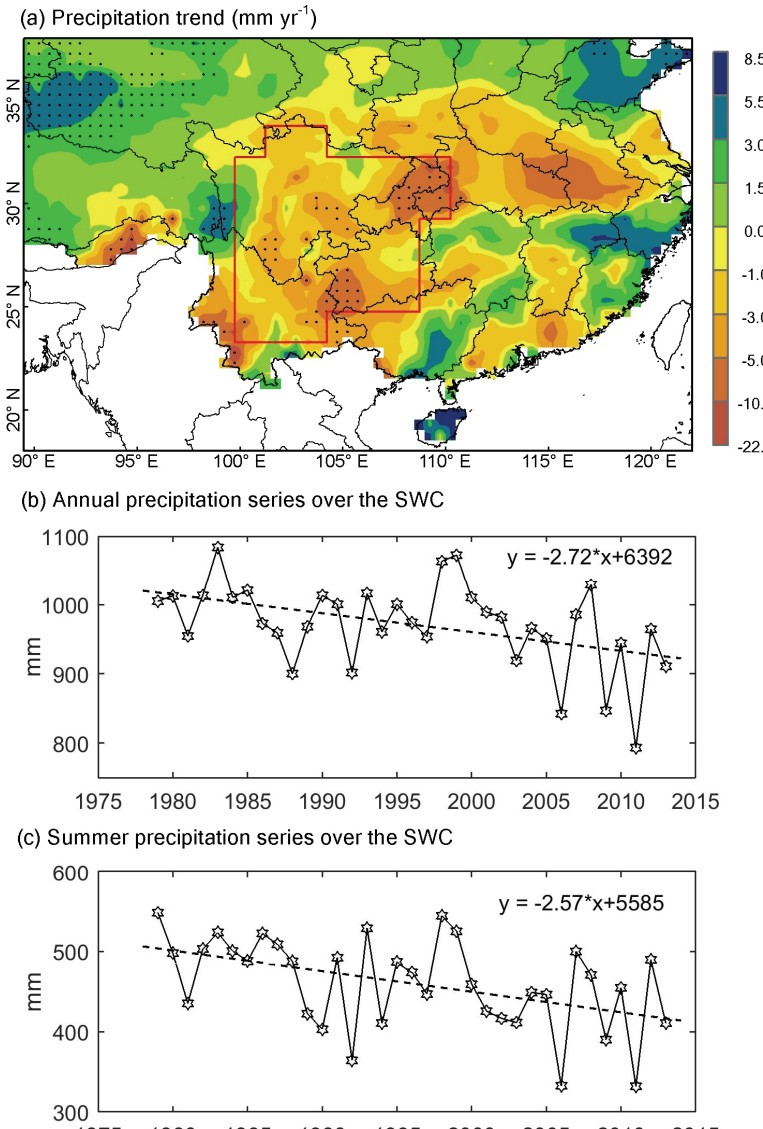

**Figure 1. (a) The trend of annual precipitation in South China from 1979 to 2013 with the CMA precipitation. The dot indicates a detected trend at the 5% statistical significance level according to the two-tailed Student's t-test. The red box represents the study**





area of SWC. The time series of (b) annual and (c) summer (July to September) SWC precipitation from 1979 to 2013. Both of the trends are significant at the 5% level based on the two-tailed Student's t-test.





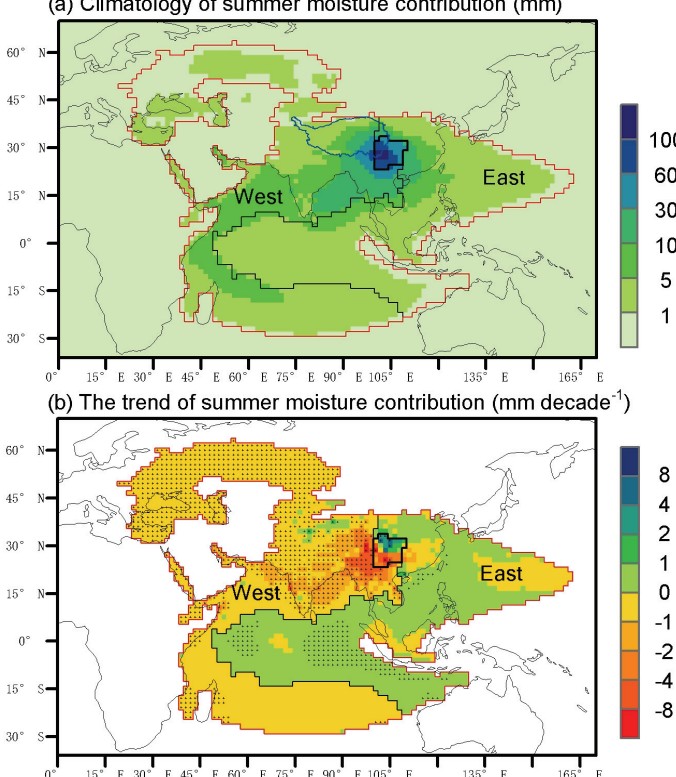

**Figure 2. (a) Climatology of the summer moisture contribution to the SWC precipitation from 1979 to 2013. The red line delineates the major source region, i.e., grids with value above 0.8 mm yr[-1]. The black line divides the major source region into the extended west, SWC, and extended east regions. The blue line delineates the Tibetan Plateau (_Zhang et al._, 2014). (b) The trend of**

5 **the summer moisture contribution from 1979 to 2013. The dots indicate a trend at the 5% significance level based on the t test. Values outside the major region are not shown.**



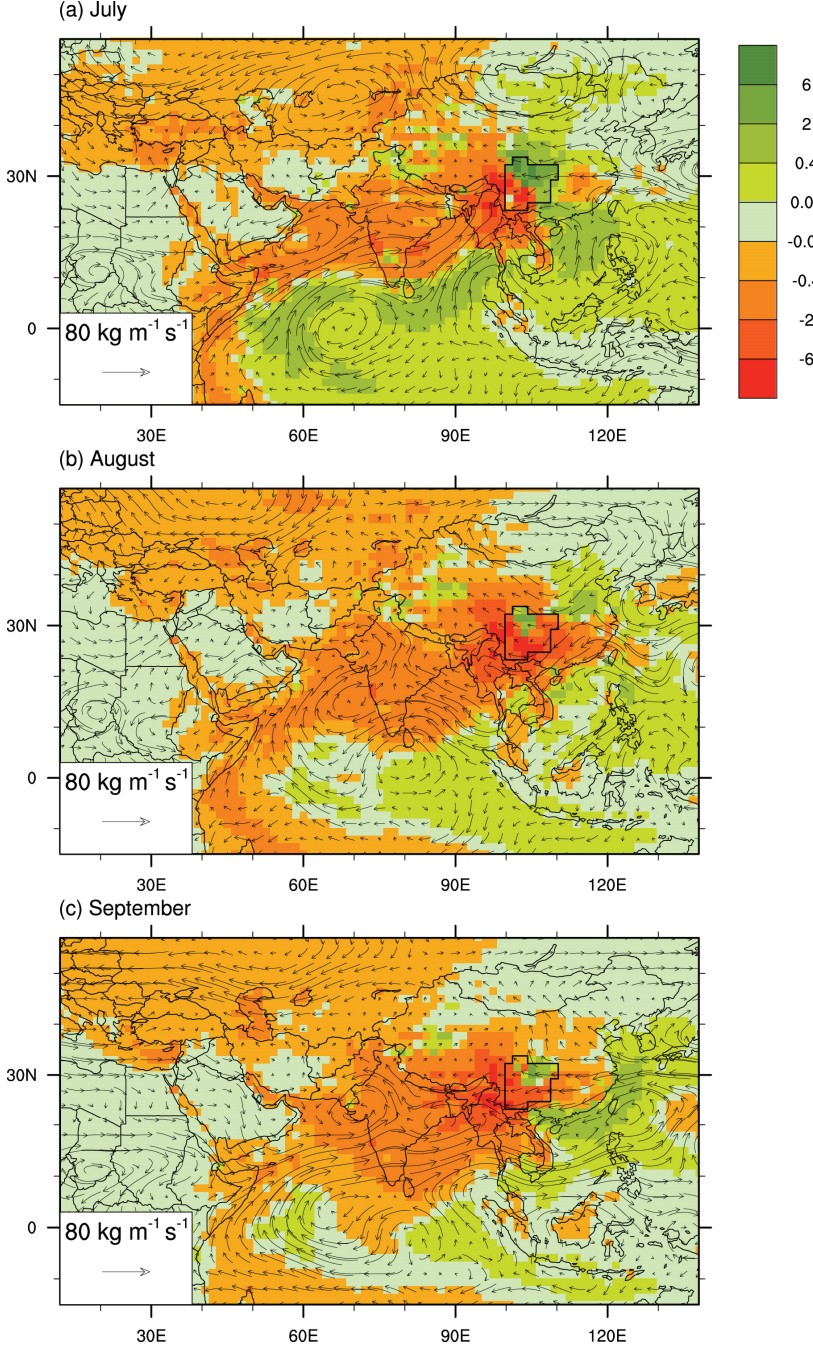





**Figure 3. The difference of July (a), August (b), and September (c) moisture contribution (unit: mm mon$^{-1}$; shading) between the last and first 10 years (last – first) of 1979–2013. The vectors represent the difference of moisture transport.**




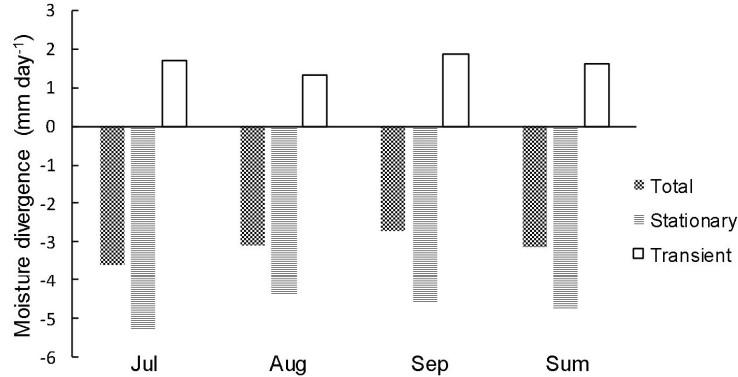

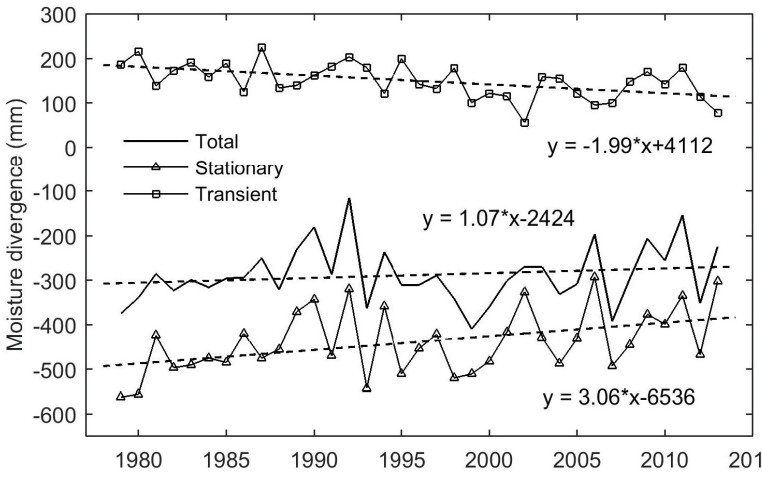

Figure 4. (a) Areal moisture divergence and its stationary and transient components over the SWC (unit: mm day⁻¹) for July, August, September, and summer during 1979–2013. (b) Annual summer moisture divergence and its stationary and transient components (unit: mm) over the SWC during 1979–2013.





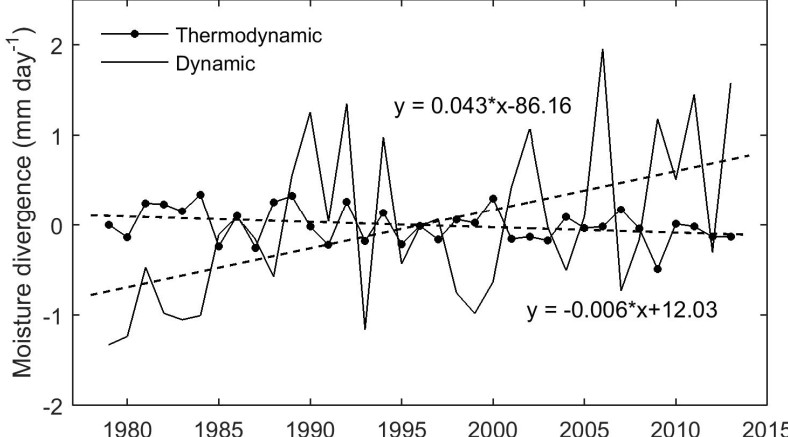

**Figure 5. The anomalies of areal summer (from July to September) moisture divergence over the SWC (unit: mm day⁻¹) caused by thermodynamic and dynamic terms during 1979–2013.**





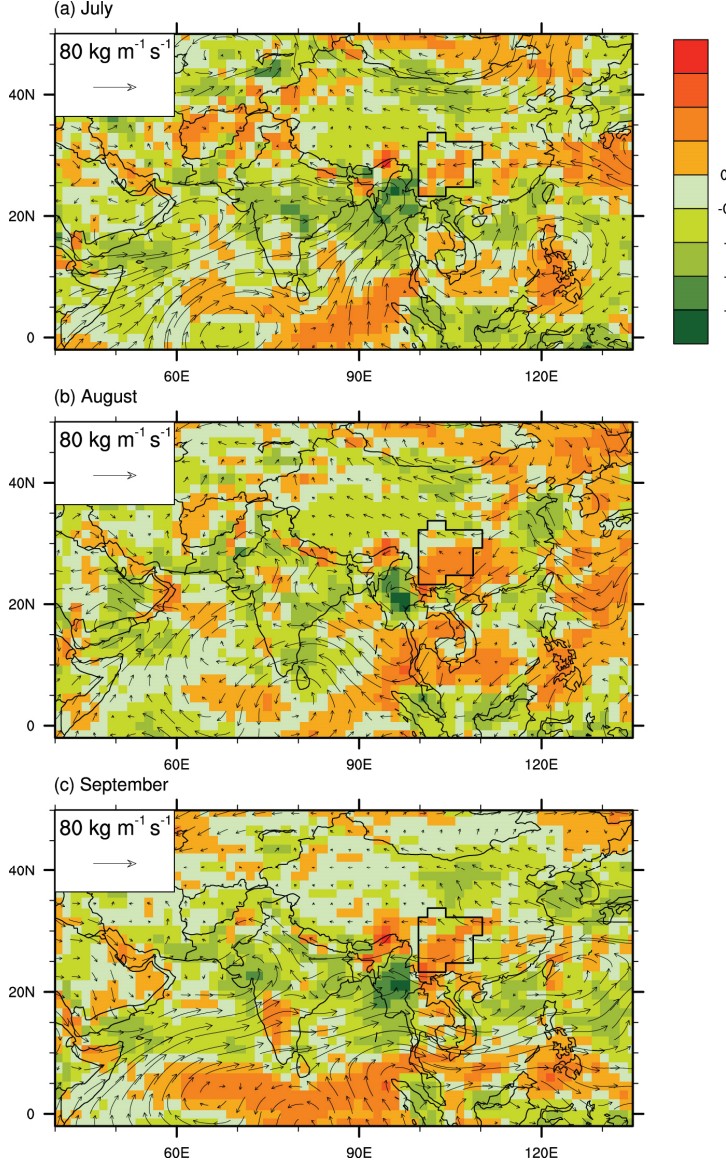

**Figure 6. The monthly difference flux (vectors) of the dynamic component of the stationary moisture transport between the last and first 10 years (last – first) and its divergence (unit: $10^{-5}$ kg m$^{-2}$ s$^{-1}$; shading).**