# Peer review of "Tracing changes in atmospheric moisture supply to the drying Southwest China"

_Atmospheric Chemistry and Physics, 2017_

## Referee Comment (RC1) · Anonymous Referee #1 · 24 Apr 2017

**REVIEW OF ZHANG ET AL**

**GENERAL COMMENTS**

The paper of Zhang et al investigates quantifies the trends in rainfall decrease in Southwest China and investigates whether and how atmospheric circulation plays a role. The paper is easy to read and the figures generally support the text and vice versa. The study is novel in the sense that atmospheric tracking has not often been applied to trends in precipitation, but rather for climatologies or variability studies only. That being said, I have a few concerns with the manuscript, which I hope the authors can address in a revised version:

- Units are not used consistently up to (what should be) the scientific standard. Precipitation should always be per a unit of time, thus mm mon$^{-1}$ and never just mm. Trends should always be per unit of time squared, thus mm yr$^{-2}$ or mm mon$^{-1}$ decade$^{-1}$ and never just mm yr$^{-1}$. Same holds for moisture flux divergence (or in fact any flux). The sister journal of ACP, HESS, has a good guide: http://www.hydrology-and-earth-system-sciences.net/for_authors/manuscript_preparation.html under mathematical requirements.
- The water accounting model (WAM) has received several updates since van der Ent et al. (2010), and it is not clear whether the authors use the updated version with two vertical layers (van der Ent et al., 2014), which is apparently open source now (van der Ent, 2016). This may be very relevant due to the wind shear present in the area under investigation, which will lead to biases when vertically integrated fluxes are being used (van der Ent et al., 2013; Goessling and Reick, 2013).
- There is limited background information on the ground-based precipitation dataset from CMA. It is always tricky to do trend analysis on interpolated data for which the stations on which the dataset is based might not be homogeneous. I suggest the authors give more information on the number of stations used, whether that is constant, are there data gaps, is it just stations or satellite information as well? And a reason why they think it is safe to apply trend analysis on this dataset.
- The decomposition of moisture transport is not well enough explained. The results seem relevant, but from the information in the paper I do not see how this could be easily reproduced.

**SPECIFIC COMMENTS**

P1,L19: "at a rate of -23.6 mm$^{-1}$ decade"
This is just one of the many examples what I mean with the wrong use of units. Because the unit is incorrect it leaves the reader wondering whether this is -23.6 mm per year per decade or -23.6 mm per decade per decade or -23.6 mm per month per decade or -23.6 mm per day per decade. Admittedly, these mistakes can be found abundantly in the scientific literature, but it is no excuse, in my opinion, to take such issues lightly, rather I hope that the authors agree with me and start correcting themselves as well as others.

P2,L27-28: "The ERA-I data have a spatial resolution of 1.5° × 1.5° grid cell"
Apparently this is the resolution that the authors used (which is ok), but other (higher) resolution are also available, thus please rephrase this sentence.

P2,L30-P3,L2: Here, the authors explain that they have replaced the evaporation and precipitation fields from ERA-I with CMA precipitation and GLDAS evaporation, because of existing "limitations in the reanalysis estimates". The claim about limitations is, however, not being backed up with a reference or figures and nor is any proof given that the alternative datasets are any better. I suggest the authors to back up this choice of data better.

P3,L5: "backward in time"

As far as I know backward tracking with WAM has been applied by Keys et al. (2012) for the first time.

P3-P4: "Section 2.3 Decomposing moisture transport"

This entire section could benefit from equations and figures to explain the concept behind decomposition.

P4,L21-22: "As shown in Fig. 2, the farther away from the target region, the lower intensity of moisture is contributed to the target (Zhang C. et al., 2017)"

I think it is a bit misplaced to cite just an own paper here as there are literally dozens of other papers that used back-trajectory methods which have found this. Moreover, it is not even as simple as put here, because it naturally depends on the winds (otherwise we could just draws circles around the target region)

P4-P5, "Section 3.1 Moisture origin"

I think previous literature is not sufficiently cited in relation to the findings of this paper. A few papers that have source region figures for China or sub-regions of China that for example could be of interest (Keys et al., 2014; Wei et al., 2012, 2016).

P7,L5: "the Asian monsoon regions"

Which are exactly? Would it perhaps make sense to delineate them somewhere?

P7: "Data availability"

What about the data availability of the CMA product? This section should be expanded according to the ACP guidelines: http://www.atmospheric-chemistry-and-physics.net/about/data_policy.html

The summer months appear to be July, August and September, whereas the meteorological summer for the northern hemisphere is generally regarded as June, July, August. Why the difference? The fact that JAS is considered should be 100% clear in all figure and table captions.

Figure 2: the caption should include what the contribution to total precipitation the red boundary in Fig. 2a encompasses. I saw it mentioned in the text, but not in the figure caption itself.

Figure 2: why is the Tibetan Plateau relevant?

Figure 2: There are multiple black lines (also the target region), which makes the caption confusing.

Fig. 2b: The information between 0 and 1 and -1 and 0 seems quite relevant, could the authors add more colors?

Figure 2: Is the boundary between East and West expert judgement? The art of the modeler? Or is there some physical determining factor?

Figure S1: What do the colors mean? The color scale lacks units or explanation in the caption.

**TECHNICAL CORRECTIONS**

As mentioned before, units should be corrected throughout the paper.

**REFERENCES**

Goessling, H. F. and Reick, C. H.: On the "well-mixed" assumption and numerical 2-D tracing of atmospheric moisture, Atmos. Chem. Phys., 13(11), 5567–5585, doi:10.5194/acp-13-5567-2013, 2013.

Keys, P. W., van der Ent, R. J., Gordon, L. J., Hoff, H., Nikoli, R. and Savenije, H. H. G.: Analyzing precipitationsheds to understand the vulnerability of rainfall dependent regions, Biogeosciences, 9(2), 733–746, doi:10.5194/bg-9-733-2012,

2012.

Keys, P. W., Barnes, E. A., van der Ent, R. J. and Gordon, L. J.: Variability of moisture recycling using a precipitationshed framework, Hydrol. Earth Syst. Sci., 18(10), 3937–3950, doi:10.5194/hess-18-3937-2014, 2014.

van der Ent, R. J.: WAM2layersPython, [online] Available from: https://github.com/ruudvdent/WAM2layersPython (Accessed 21 April 2017), 2016.

van der Ent, R. J., Savenije, H. H. G., Schaefli, B. and Steele-Dunne, S. C.: Origin and fate of atmospheric moisture over continents, Water Resour. Res., 46(9), W09525, doi:10.1029/2010WR009127, 2010.

van der Ent, R. J., Tuinenburg, O. A., Knoche, H. R., Kunstmann, H. and Savenije, H. H. G.: Should we use a simple or complex model for moisture recycling and atmospheric moisture tracking?, Hydrol. Earth Syst. Sci., 17(12), 4869–4884, doi:10.5194/hess-17-4869-2013, 2013.

van der Ent, R. J., Wang-Erlandsson, L., Keys, P. W. and Savenije, H. H. G.: Contrasting roles of interception and transpiration in the hydrological cycle - Part 2: Moisture recycling, Earth Syst. Dyn., 5(2), 471–489, doi:10.5194/esd-5-471-2014, 2014.

Wei, J., Dirmeyer, P. A., Bosilovich, M. G. and Wu, R.: Water vapor sources for Yangtze River Valley rainfall: Climatology, variability, and implications for rainfall forecasting, J. Geophys. Res., 117(5), 1–11, doi:10.1029/2011JD016902, 2012.

Wei, J., Knoche, H. R. and Kunstmann, H.: Atmospheric residence times from transpiration and evaporation to precipitation: An age-weighted regional evaporation tagging approach, J. Geophys. Res. Atmos., 121, 6841–6862, doi:10.1002/2015JD024650, 2016.

---

## Referee Comment (RC2) · Anonymous Referee #2 · 19 May 2017

Review of "Tracing changes in atmospheric moisture supply to the drying Southwest China" by Zhang et al., submitted to ACPD

The authors present a climatological study of moisture transport to a region in China, analysing trends in precipitation from Reanalysis and gridded station data. Trends in moisture origin and moisture convergence are also analysed. The paper is generally well-written and the methods are mostly sound. However, there are several uncertainties in this analysis that need further discussion, and some statements need aditional justification. I suggest to include another section providing a discussion of the method uncertainties and the possible impact on the results. Furthermore, I could imagine that the manuscript may possibly find more readership in a specific climate journal, maybe the Handling Co-Editor has some thoughts on this. Below I detail my major and specific comments.

[Figure]

Major comments

1. The method description should be improved by explicitly naming some of the uncerlying assumptions. For example, the calculation of a proportion of precipitation relative to total column water implies that water vapour is well-mixed at every grid cell. Do the results depend on grid spacing of the input data? A figure would help to support the explanation of how the WAM method works.

2. The authors state in Sec. 2.2 that after 30 days, "a large amount of water may be left in the air" and that they continue for another 30 days. Please quantify how large an amount is left in the atmosphere after 30 days. What is the origin of this uncertainty? How realistic is it to assume precipitation water stays in the atmosphere for 60 days (2 months) in this region?

3. The method uses a combination of reanalysis and observational data that are blended together and partly rescaled. Could it be that inconsistencies between the ERA-Interim water cycle and observations bias the results, and impact the trend analysis? What are the uncertainties of this combination of data used? How do uncertainties in P and E parameterisations influence the results? This should be discussed and evaluated in more detail, maybe in a separate "method sensitivity/method discussion" section.

4. The display of moisture contributions in Fig. 2 is cut off at 0.8 mm, providing 88.3% of the total precipitation. Is there a justification for choosing this percentile? It would be helpful to add also the contours encompassing 50% and 95% of the total precipitation in the figure.

5. The trend obtained in the analyses seem to depend strongly on the years 2006 and 2011. Is there a significant trend observed if these two years were removed from the time series? How reliable are trends from reanalysis data in general?

6. Some conclusions seem not sufficiently based on evidence in the manuscript. This

includes the statement that "local recycling played a minor role" (pg. 5, L. 12), that the "dominant role of dynamic process ... prevails over a very large area" (pg. 6, L. 20), and the speculation of a possible role of SST anomalies in the changes (pg. 6, L. 30). Notably, the "might be related" in that line becomes a "likely related" in the conclusions (pg. 7, L. 16), even though no evidence to that end is presented in the manuscript. A clearer and more balanced argumentation, including alternative interpretations, should be formulated in all of these cases. The statement "the westerlies play a secondary role" (pg. 7, L. 5) has also no clear anchoring the the results presented before.

7. More references to the literature on the topic of moisture source analyses should be included in the introduction and in the discussion of the uncertainties of the results. In addition to the studies by Gustafsson and Zhang, consider some of the earlier founding work from Stohl and James (2004,2005), James et al. (2004), Sodemann et al. (2008), Sodemann and Zubler (2010), Baker et al. (2015), Winschall et al. (2014). Also relevant are the discussion of the uncertainties of the well-mixed assumption (Goessling and Reick, 2013).

Specific comments

Pg. 1, L. 14: "monsoon region" please specify which monsoon region

Pg. 2, L. 4: descend flows -> descending motion

Pg. 2, L. 21: seems somewhat circular, please rephrase. Analysing the moisture sources and transport appears to me as another way of looking at circulation patterns, but with a focus on one aspect of precipitation (the other one being lifing/condensation).

Pg. 2, L. 27: Unclear how the Trenberth (1991) citation fits in here.

Pg. 2, L. 28: grid cell -> degree

Pg. 3, L. 8: Is this vertially integrated moisture transport?

Pg. 3, L. 25: Please clarify how exactly the rescaling was done in order to ensure

reproducibility of your results. What rescaling factor was used?

Pg. 4, L. 5: At what level where q and wind velocity considered for this analysis?

Pg. 4, L. 10: has experienced -> shows

Pg. 4, L. 22-23: lapses -> decrases

Pg. 4, L. 24: delete "precipitation"

Pg. 4, L. 24: the finding that humid regions provide more moisture than arid regions is quite obvious; the description could provide more quantitative detail

Pg. 5, L. 8: Figure 2 has already been introduced above

Pg. 5, L. 15: please define what you mean by "moisture supply"

Pg. 5, L. 15-25: How dependent are these results on the threshold of 0.8mm? In general, I find the moisture flux change vectors difficult to relate to the moisture contribution change, because the moisture flux is calculated for the entire atmospheric humidity, and not for the contribution to the target region.

Pg. 5, L. 30: how was moisture divergence calculated?

Pg. 5, L. 32: "the close correlation": is that the only possible conclusion? My understanding is that moisture divergence is related to precipitation by mass balance requirements, but does not provide insight into the roles of moisture transport vs. local evaporation. Please elaborate.

Pg. 6, L. 12: Obvious is a quite subjective term. Are the trends significant? How reliable are such trends from reanalysis data?

Pg. 6, L. 20: Data availability for the CMA data should be stated.

Figure 1a: Please provide a wider area in the figure panel, including some topography contours and maybe country names for orientation. A distinction between the national boundaries and province boundaries would also be helpful.

Figure 2a: Does the green shading indicate that all areas shown in the figure panel contribute >0 mm to the target area?

Figure 3: Is it possible to restrict the shading and moisture flux vectors to moisture arriving in the target region only?

References

A. Baker, H. Sodemann, J. U. L. Baldini, S. F. M. Breitenbach, K. R. Johnson, J. van Hunen, and Z. Pingzhong, 2015: Seasonality of westerly moisture transport in the East Asian Summer Monsoon and its implications for interpreting precipitation d18O, J. Geophys. Res., 120, Âădoi:10.1002/2014JD022919.

H. F. Goessling and C. H. Reick, On the 'well-mixed' assumption and numerical 2-D tracing of atmospheric moisture, Atmos. Chem. Phys., vol. 13, no. 11, pp. 5567–5585, 2013.

P. James, A. Stohl, N. Spichtinger, S. Eckhardt, and C. Forster, Climatological aspects of the extreme European rainfall of August 2002 and a trajectory method for estimating the associated evaporative source regions, Nat. Hazards Earth Syst. Sci., vol. 4, no. 5, pp. 733–746, 2004.

H. Sodemann, C. Schwierz, and H. Wernli, Interannual variability of Greenland winter precipitation sources: Lagrangian moisture diagnostic and North Atlantic Oscillation influence, J. Geophys. Res, vol. 113, no. 3, p. D03107, Feb. 2008.

H. Sodemann and E. Zubler, Seasonal and inter-annual variability of the moisture sources for Alpine precipitation during 1995-2002, Int. J. Climatol., vol. 30, no. 5, pp. 947–961, 2010.

A. Stohl and P. James, A Lagrangian analysis of the atmospheric branch of the global water cycle. Part II: Moisture transports between Earth's ocean basins and river catchments, J. Hydrometeorol., vol. 6, no. 12, pp. 961–984, 2005.

A. Stohl and P. James, A Lagrangian analysis of the atmospheric branch of the global water cycle. Part I: Method description, validation, and demonstration for the August 2002 flooding in Central Europe, J. Hydrometeorol., vol. 5, no. 8, pp. 656–678, 2004.

A. Winschall, S. Pfahl, H. Sodemann, and H. Wernli, Comparison of Eulerian and Lagrangian moisture source diagnostics - the flood event in eastern Europe in May 2010, Atmos. Chem. Phys., vol. 14, no. 13, pp. 6605–6619, 2014.
* * *

---

## Author Comment (AC1)

Reviewer #1

The paper of Zhang et al investigates quantifies the trends in rainfall decrease in Southwest China and investigates whether and how atmospheric circulation plays a role. The paper is easy to read and the figures generally support the text and vice versa. The study is novel in the sense that atmospheric tracking has not often been applied to trends in precipitation, but rather for climatologies or variability studies only.

*A: Thanks for the comments. We truly appreciate it.*

That being said, I have a few concerns with the manuscript, which I hope the authors can address in a revised version:

1. Units are not used consistently up to (what should be) the scientific standard. Precipitation should always be per a unit of time, thus mm mon$^{-1}$ and never just mm. Trends should always be per unit of time squared, thus mm yr$^{-2}$ or mm mon$^{-1}$ decade$^{-1}$ and never just mm yr$^{-1}$. Same holds for moisture flux divergence (or in fact any flux). The sister journal of ACP, HESS, has a good guide: http://www.hydrology-and-earth-system-sciences.net/for_authors/manuscript_preparation.html under mathematical requirements.

*A: Units have been changed to fit the standard of ACP in the revision. The trends are expressed as per unit of time squared. The fluxes including precipitation and moisture divergence are all expressed as per unit of time throughout the paper.*

2. The water accounting model (WAM) has received several updates since van der Ent et al. (2010), and it is not clear whether the authors use the updated version with two vertical layers (van der Ent et al., 2014), which is apparently open source now (van der Ent, 2016). This may be very relevant due to the wind shear present in the area under investigation, which will lead to biases when vertically integrated fluxes are being used (van der Ent et al., 2013; Goessling and Reick, 2013).

*A: The original version of WAM (WAM1), instead of the two vertical layer model (WAM2), was applied in this study. To address the concern of the reviewer about model versions, we tested the potential contribution of wind shear to moisture flux by*

*calculating the moisture flux shear factor following Eqs. (7-8) in van der Ent et al. (2013) in the target area (see Fig. r1). Lower value indicates stronger moisture flux shear. The areal-weighted shear factor at the zonal direction over summer (JAS) in study area SWC during 1979-2013 is 0.72, while it is 0.76 at the meridional direction. Taking the zonal shear factor as an example, a shear factor of 0.72 means that 86% of the water goes in one direction with 14% in the opposite direction. Because the dominant moisture flux has a high share of the overall flux, the moisture flux shear is not strong in this case.*

*To further confirm that, we selected the year 1986 with the strongest moisture flux shear (The averaged zonal and meridional shear factor in summer 1986 is 0.71.) as a sample case and applied the WAM2 for comparison. The results (Fig. r2) show that the spatial patterns of moisture contribution match with each other between WAM1 and WAM2. This is not the case as in van der Ent et al. (2013), where WAM2 shew distinct spatial pattern with WAM1. Thus, WAM1 is considered suitable for this study. We have added the comparison of WAM versions into the discussion section.*

[Figure]

*Fig. r1 Horizontal moisture flux shear factor in summer (JAS) averaged over 1979–2013 with ERA-I. (a) Zonal moisture flux shear factor, (b) meridional moisture flux shear factor.*

[Figure]

*Fig. r2 Moisture contribution for JAS precipitation in 1986 SWC with WAM1 (a) and WAM2 (b).*

3. There is limited background information on the ground-based precipitation dataset from CMA. It is always tricky to do trend analysis on interpolated data for which the stations on which the dataset is based might not be homogeneous. I suggest the authors give more information on the number of stations used, whether that is constant, are there data gaps, is it just stations or satellite information as well? And a reason why they think it is safe to apply trend analysis on this dataset.

*A: The CMA (China Meteorological Administration) precipitation dataset is based on ground observations of ~2400 stations over China (Shen and Xiong, 2016). The stations within this study area are shown in Fig. r3. The data are quality controlled and released*

*by CMA. It is seen as one of the best ground-based precipitation products over China and it is widely used in many studies. In the revision, a brief discussion on the CMA dataset is provided.*

[Figure]

*Fig. r3 Rain stations distribution. The circle represents the 0.5° grid with at least one rain station.*

4. The decomposition of moisture transport is not well enough explained. The results seem relevant, but from the information in the paper I do not see how this could be easily reproduced.

*A: Section "2.3 Decomposing moisture transport" has been revised by adding formulas to explain the decomposition of moisture transport. We have added a short paragraph to detail the analysis method. The decomposition process was described in detail in the references Li et al. (2013) and Seager et al. (2010). As it is a widely used procedure in the research field, it should not be difficult to reproduce following the guidance in the references.*

5. P1, L19: "at a rate of -23.6 mm$^{-1}$ decade" This is just one of the many examples what I mean with the wrong use of units. Because the unit is incorrect it leaves the reader wondering whether this is -23.6 mm per year per decade or -23.6 mm per decade per decade or -23.6 mm per month per decade or -23.6 mm per day per decade. Admittedly,

these mistakes can be found abundantly in the scientific literature, but it is no excuse, in my opinion, to take such issues lightly, rather I hope that the authors agree with me and start correcting themselves as well as others.

*A: We agree with the reviewer on the unit issue. The units have been carefully checked and corrected in the revision.*

6. P2, L27-28: "The ERA-I data have a spatial resolution of 1.5° × 1.5° grid cell" Apparently this is the resolution that the authors used (which is ok), but other (higher) resolution are also available, thus please rephrase this sentence.

*A: There is ERA-I data with other resolutions, and the 1.5 degree data were used in this study. The sentence has been rephrased.*

7. P2, L30-P3, L2: Here, the authors explain that they have replaced the evaporation and precipitation fields from ERA-I with CMA precipitation and GLDAS evaporation, because of existing "limitations in the reanalysis estimates". The claim about limitations is, however, not being backed up with a reference or figures and nor is any proof given that the alternative datasets are any better. I suggest the authors to back up this choice of data better.

*A: It is well-known that the reanalysis data such as ERA-I have large biases on precipitation estimates, and so as the evaporation estimates (Trenberth et al., 2011; Tong et al., 2013). The CMA precipitation data is from ground-based observations and was released by CMA, the Administration which is responsible for meteorological observations in China (http://www.cma.gov.cn/en2014/). The GLDAS is forced with precipitation gauge observations among others (Rodell et al., 2004). Both datasets are observation-based and are better than the estimates in the reanalysis data. In the revision, we have provided the related references to back up the choice of data.*

8. P3, L5: "backward in time" As far as I know backward tracking with WAM has been applied by Keys et al. (2012) for the first time.

*A: We have corrected the citation and have added Keys et al. (2012) in the revision.*

9. P3-P4: "Section 2.3 Decomposing moisture transport" This entire section could benefit from equations and figures to explain the concept behind decomposition.

*A: This section has been revised by adding formulas to explain the decomposition method.*

10. P4, L21-22: "As shown in Fig. 2, the farther away from the target region, the lower intensity of moisture is contributed to the target (Zhang C. et al., 2017)"
I think it is a bit misplaced to cite just an own paper here as there are literally dozens of other papers that used back-trajectory methods which have found this. Moreover, it is not even as simple as put here, because it naturally depends on the winds (otherwise we could just draws circles around the target region)

*A: Thanks for your suggestions. More references have been added in the revision. In addition, the statement has been modified to reflect the key message, that is, the moisture contribution differs in different directions depends on the winds. As shown in the Fig. 2, for the study area, more moisture is from the south and less moisture from the north.*

11. P4-P5, "Section 3.1 Moisture origin" I think previous literature is not sufficiently cited in relation to the findings of this paper. A few papers that have source region figures for China or sub-regions of China that for example could be of interest (Keys et al., 2014; Wei et al., 2012, 2016).

*A: We have cited the references and compared the findings in the previous references and those in this study. The comparisons are generally consistent, although the focus areas are not the same. We have also emphasized that we focus more on the changes of the moisture contribution, rather than the climatological pattern in this study.*

12. P7, L5: "the Asian monsoon regions" Which are exactly? Would it perhaps make sense to delineate them somewhere?

*A: "The Asian monsoon regions" here means from the northern Indian Ocean to SWC*

*and from South China Sea to SWC. We have modified the statement and specified the exact regions in the revision.*

13. P7: "Data availability" What about the data availability of the CMA product? This section should be expanded according to the ACP guidelines: http://www.atmospheric-chemistry-and-physics.net/about/data_policy.html

*A: The CMA product was released by CMA and was downloaded from the China Meteorological Data Service Center (CMDC, http://data.cma.cn/en, http://data.cma.cn/data/cdcdetail/dataCode/SURF_CLI_CHN_PRE_DAY_GRID_0.5.html). The site has been provided in the Data availability section in the revision.*

14. The summer months appear to be July, August and September, whereas the meteorological summer for the northern hemisphere is generally regarded as June, July, August. Why the difference? The fact that JAS is considered should be 100% clear in all figure and table captions.

*A: We focused on July, August and September because precipitation in these months shows large and significant trends over the study area. We agree that meteorological summer usually indicates June, July, and August. In the revision, we have added a note on the specification of "summer" in this study, The JAS represented summer is clarified throughout the paper.*

15. Figure 2: the caption should include what the contribution to total precipitation the red boundary in Fig. 2a encompasses. I saw it mentioned in the text, but not in the figure caption itself.

*A: The caption has been revised to include the information.*

16. Figure 2: why is the Tibetan Plateau relevant?

*A: The Tibetan Plateau situates in the upwind of SWC. Due to its high altitude, it is well-known that it will block the moisture from the west to SWC (Tian et al., 2007; Yu et al., 2008). It can be seen from Figure 2 that little moisture contribution from the west*

*of the Tibetan Plateau. However, the Tibetan Plateau itself may contribute moisture to SWC (Huang and Cui, 2015), which is also seen from Figure 2. Thus, it explains the moisture contribution pattern (relative high contribution from eastern Tibetan Plateau, but close to zero contribution from the western and west to the Tibetan Plateau) shown in Figure 2. It is an important finding regarding how the high altitude of the Tibetan Plateau affects moisture contribution pattern. These reasons have been clarified in the revision.*

17. Figure 2: There are multiple black lines (also the target region), which makes the caption confusing.

*A: The line colors have been adjusted. The target region is changed to brown. The land outline is changed to gray. The division line between East and West is black.*

18. Fig. 2b: The information between 0 and 1 and -1 and 0 seems quite relevant, could the authors add more colors?

*A: The color bar has been modified following the suggestion.*

19. Figure 2: Is the boundary between East and West expert judgement? The art of the modeler? Or is there some physical determining factor?

*A: The boundary is based on the result (Fig. 2b) following the distinct opposite sign of the moisture trend. In the west side, the moisture contribution is decreasing while it is increasing in the east side. It has been clarified in the revision.*

20. Figure S1: What do the colors mean? The color scale lacks units or explanation in the caption.

*A: The colors represent the moisture divergence of the fluxes. The units have been added in the caption on the next page.*

21. TECHNICAL CORRECTIONS
As mentioned before, units should be corrected throughout the paper.

*A: The units have been corrected throughout the revised paper.*

**References**

Huang, Y., and Cui, X.: Moisture Sources of Torrential Rainfall Events in the Sichuan Basin of China during Summers of 2009–13, J. Hydrometeor., 16, 1906–1917, doi: 10.1175/JHM-D-14-0220.1, 2015.

Li, L., Li, W., and Barros, A. P.: Atmospheric moisture budget and its regulation of the summer precipitation variability over the Southeastern United States., Clim. Dyn., 41(3-4), 613-631, 2013.

Rodell, M., Houser, P., Jambor, U., Gottschalck, J., Mitchell, K., Meng, C., Arsenault, K., Cosgrove, B., Radakovich, J., Bosilovich, M., Entin, J., Walker, J., Lohmann, D., and Toll D.: The Global Land Data Assimilation System, Bull. Amer. Meteor. Soc., 85, 381–394, 2004.

Seager, R., Naik, N., Vecchi, G. A.: Thermodynamic and dynamic mechanisms for large-scale changes in the hydrological cycle in response to global warming. J Clim 23:4651–4668, 2010.

Shen, Y., and Xiong, A.: Validation and comparison of a new gauge-based precipitation analysis over mainland China. Int. J. Climatol., 36(1), 252-265, 2016.

Tian, L., Yao, T., MacClune, K., White, J. W. C., Schilla, A., Vaughn, B., Vachon, R., and Ichiyanagi, K. Stable isotopic variations in west China: a consideration of moisture sources. J. Geophys. Res. Atmos., 112(D10), 2007.

Tong, K., Su, F., Yang, D., Zhang, L., and Hao, Z.: Tibetan Plateau precipitation as depicted by gauge observations, reanalyses and satellite retrievals. Int. J. Climatol., 34(2), 265-285, 2014.

van der Ent, R. J., Tuinenburg, O. A., Knoche, H. R., Kunstmann, H. and Savenije, H. H. G.: Should we use a simple or complex model for moisture recycling and atmospheric moisture tracking? Hydrol. Earth Syst. Sci., 17(12), 4869–4884, doi:10.5194/hess-17-4869-2013, 2013.

Yu, W., Yao, T., Tian, L., Ma, Y., Ichiyanagi, K., Wang, Y., and Sun, W.: Relationships between $\delta^{18}O$ in precipitation and air temperature and moisture origin on a south–north transect of the Tibetan Plateau. Atmos. Res., 87(2), 158-169, 2008.

---

## Author Response (AR1)

We thank the two anonymous reviewers for the constructive comments and appreciate their expertise. Here are the point-to-point responses to their comments.

**Reviewer #1**

The paper of Zhang et al investigates quantifies the trends in rainfall decrease in Southwest China and investigates whether and how atmospheric circulation plays a role. The paper is easy to read and the figures generally support the text and vice versa. The study is novel in the sense that atmospheric tracking has not often been applied to trends in precipitation, but rather for climatologies or variability studies only.

**A: Thanks for the comments. We truly appreciate it.**

That being said, I have a few concerns with the manuscript, which I hope the authors can address in a revised version:

1. Units are not used consistently up to (what should be) the scientific standard. Precipitation should always be per a unit of time, thus mm mon-1 and never just mm. Trends should always be per unit of time squared, thus mm yr-2 or mm mon-1 decade-1 and never just mm yr-1. Same holds for moisture flux divergence (or in fact any flux). The sister journal of ACP, HESS. has а good guide: http://www.hydrology-and-earth-system-sciences.net/for authors/manuscript prepara tion.html under mathematical requirements.

*A*: Units have been changed to fit the standard of *ACP* in the revision. The trends are expressed as per unit of time squared. The fluxes including precipitation and moisture divergence are all expressed as per unit of time throughout the paper.

2. The water accounting model (WAM) has received several updates since van der Ent et al. (2010), and it is not clear whether the authors use the updated version with two vertical layers (van der Ent et al., 2014), which is apparently open source now (van der Ent, 2016). This may be very relevant due to the wind shear present in the area under investigation, which will lead to biases when vertically integrated fluxes are

being used (van der Ent et al., 2013; Goessling and Reick, 2013).

A: The original version of WAM (WAM1), instead of the two vertical layer model (WAM2), was applied in this study. To address the concern on model versions, we tested the potential contribution of wind shear to moisture flux by calculating the moisture flux shear factor following Eqs. (7-8) in van der Ent et al. (2013) in the target area (see Fig. 8 in revision). Lower value indicates stronger moisture flux shear. The areal-weighted shear factor at the zonal direction over summer (JAS) in study area SWC during 1979-2013 is 0.72, while it is 0.76 at the meridional direction. Taking the zonal shear factor as an example, a shear factor of 0.72 means that 86% of the water goes in one direction with 14% in the opposite direction. Because the dominant moisture flux has a high share of the overall flux, the moisture flux shear is not strong in this case.

To further confirm that, we selected the year 1986 with the strongest moisture flux shear (The averaged zonal and meridional shear factor in summer 1986 is 0.71) as a sample case and applied the WAM2 for comparison. The results (Fig. 9) show that the spatial patterns of moisture contribution match with each other between WAM1 and WAM2. This is not the case as in van der Ent et al. (2013), where WAM2 shew distinct spatial pattern with WAM1. Thus, WAM1 is considered suitable for this study. We have added the results of moisture flux shear and the comparison of two WAM versions into the 4.1 section of sensitivity analysis on WAM.

3. There is limited background information on the ground-based precipitation dataset from CMA. It is always tricky to do trend analysis on interpolated data for which the stations on which the dataset is based might not be homogeneous. I suggest the authors give more information on the number of stations used, whether that is constant, are there data gaps, is it just stations or satellite information as well? And a reason why they think it is safe to apply trend analysis on this dataset.

A: The CMA (China Meteorological Administration) precipitation dataset is based on ground observations of ~2400 stations over China (Shen and Xiong, 2016). The stations within this study area are shown in revised Fig. 1b. The data are quality controlled and released by CMA. It is seen as one of the best ground-based precipitation products over China and it is widely used in many studies. In the revision, a brief discussion on the CMA dataset is provided (P3, L5-7).

4. The decomposition of moisture transport is not well enough explained. The results seem relevant, but from the information in the paper I do not see how this could be easily reproduced.

A: Section "2.3 Decomposing moisture transport" has been revised by adding formulas to explain the decomposition of moisture transport. We have added short subparagraphs to detail the analysis method (P4, L25-28 and P5, L4-7). The decomposition process was described in detail in the references Li et al. (2013) and Seager et al. (2010). As it is a widely used procedure in the research field, it should not be difficult to reproduce following the guidance in the references.

5. P1, L19: "at a rate of -23.6 mm-1 decade" This is just one of the many examples what I mean with the wrong use of units. Because the unit is incorrect it leaves the reader wondering whether this is -23.6 mm per year per decade or -23.6 mm per decade or -23.6 mm per decade or -23.6 mm per decade. Admittedly, these mistakes can be found abundantly in the scientific literature, but it is no excuse, in my opinion, to take such issues lightly, rather I hope that the authors agree with me and start correcting themselves as well as others.

*A:* We agree with the reviewer on the unit issue. The units have been carefully checked and corrected in the revision.

6. P2, L27-28: "The ERA-I data have a spatial resolution of  $1.5^{\circ} \times 1.5^{\circ}$  grid cell" Apparently this is the resolution that the authors used (which is ok), but other (higher) resolution are also available, thus please rephrase this sentence.

*A:* There are ERA-I data with other resolutions, and the 1.5 degree data were used in this study. The sentence has been rephrased in P2,L32-P3,L1.

7. P2, L30-P3, L2: Here, the authors explain that they have replaced the evaporation and precipitation fields from ERA-I with CMA precipitation and GLDAS evaporation, because of existing "limitations in the reanalysis estimates". The claim about limitations is, however, not being backed up with a reference or figures and nor is any proof given that the alternative datasets are any better. I suggest the authors to back up this choice of data better.

A: It is well-known that the reanalysis data such as ERA-I have large biases on precipitation estimates, and so as the evaporation estimates (Trenberth et al., 2011; Tong et al., 2013). The CMA precipitation data is from ground-based observations and was released by CMA, the Administration which is responsible for meteorological observations in China (http://www.cma.gov.cn/en2014/). The GLDAS is forced with precipitation gauge observations among others (Rodell et al., 2004). Both datasets are observation-based and are better than the estimates in the reanalysis data. In the revision, we have provided the related references to back up the choice of data (P3, L3-4, L6-7, L9, respectively).

8. P3, L5: "backward in time" As far as I know backward tracking with WAM has been applied by Keys et al. (2012) for the first time.

*A:* We have corrected the citation and have added Keys et al. (2012) at P3, L18 in the revision.

9. P3-P4: "Section 2.3 Decomposing moisture transport" This entire section could benefit from equations and figures to explain the concept behind decomposition.

*A:* This section has been revised by adding formulas to explain the decomposition method.

10. P4, L21-22: "As shown in Fig. 2, the farther away from the target region, the lower intensity of moisture is contributed to the target (Zhang C. et al., 2017)"I think it is a bit misplaced to cite just an own paper here as there are literally dozens of other papers that used back-trajectory methods which have found this. Moreover, it

is not even as simple as put here, because it naturally depends on the winds (otherwise we could just draws circles around the target region)

A: Thanks for your suggestions. More references have been added in the revision (P6, L2). In addition, the statement has been modified to reflect the key message, that is, the moisture contribution differing in different directions depends on the winds (P6, L2-4). As shown in the Fig. 3a, for the study area, more moisture is from the south and less moisture from the north.

11. P4-P5, "Section 3.1 Moisture origin" I think previous literature is not sufficiently cited in relation to the findings of this paper. A few papers that have source region figures for China or sub-regions of China that for example could be of interest (Keys et al., 2014; Wei et al., 2012, 2016).

A: We have cited some references and compared the findings with the previous references in this study (P6, L5-6; L23-27). The comparisons are generally consistent, although the focus areas are not the same. We have also emphasized that we focus more on the changes of the moisture contribution, rather than the climatological pattern in this study (P1, L26-28).

12. P7, L5: "the Asian monsoon regions" Which are exactly? Would it perhaps make sense to delineate them somewhere?

A: "The Asian monsoon regions" here means from the northern Indian Ocean to SWC and from South China Sea to SWC. We have modified the statement and specified the exact regions in the revision (P10, L19-20).

13. P7: "Data availability" What about the data availability of the CMA product? This section should be expanded according to the ACP guidelines: http://www.atmospheric-chemistry-and-physics.net/about/data\_policy.html

A: The CMA product was released by CMA and was downloaded from the China Meteorological Data Service Center (CMDC, http://data.cma.cn/en, http://data.cma.cn/data/cdcdetail/dataCode/SURF\_CLI\_CHN\_PRE\_DAY\_GRID\_0.5.

*html*). The site has been provided in the Data availability section in the revision.

14. The summer months appear to be July, August and September, whereas the meteorological summer for the northern hemisphere is generally regarded as June, July, August. Why the difference? The fact that JAS is considered should be 100% clear in all figure and table captions.

A: We focused on July, August and September because precipitation in these months shows large and significant trends over the study area. We agree that meteorological summer usually indicates June, July, and August. In the revision, we have limited the use of summer, instead we use July, August and September (JAS) throughout the manuscript.

15. Figure 2: the caption should include what the contribution to total precipitation the red boundary in Fig. 2a encompasses. I saw it mentioned in the text, but not in the figure caption itself.

A: The caption has been revised to include the information (Figure 3; P18, L3-4).

**16. Figure 2: why is the Tibetan Plateau relevant?**

A: The Tibetan Plateau situates in the upwind of SWC. Due to its high altitude, it is well-known that it will block the moisture from the west to SWC (Tian et al., 2007; Yu et al., 2008). It can be seen from Figure 3a that little moisture contribution from the west of the Tibetan Plateau. However, the Tibetan Plateau itself may contribute moisture to SWC (Huang and Cui, 2015), which is also seen from Figure 3a. Thus, it explains the moisture contribution pattern (relative high contribution from eastern Tibetan Plateau, but close to zero contribution from the western and west to the Tibetan Plateau). It is an important finding regarding how the high altitude of the Tibetan Plateau affects moisture contribution pattern. These reasons have been clarified in the revision (P6, L19-24).

17. Figure 2: There are multiple black lines (also the target region), which makes the

caption confusing.

A: The line colors have been adjusted. The target region is changed to brown. The land outline is changed to gray. The division line between East and West is black. And the figure is also changed to Figure 3.

18. Fig. 2b: The information between 0 and 1 and -1 and 0 seems quite relevant, could the authors add more colors?

*A*: *The color bar has been modified following the suggestion (see Fig. 3b).*

19. Figure 2: Is the boundary between East and West expert judgement? The art of the modeler? Or is there some physical determining factor?

*A*: The boundary is based on the result (Fig. 3b) following the distinct opposite sign of the moisture trend. In the west side, the moisture contribution is decreasing while it is increasing in the east side. It has been clarified in the text (P6, L11-12).

20. Figure S1: What do the colors mean? The color scale lacks units or explanation in the caption.

*A*: The colors represent the moisture divergence of the fluxes. The units have been added in the caption on the next page (P2, L2-4).

**21. TECHNICAL CORRECTIONS**

As mentioned before, units should be corrected throughout the paper.

*A*: *The units have been corrected throughout the revised paper.*

- 15 Indian Ocean to SWC and from South China Sea to SWC. The major moisture contributing area is divided into an extended west region, SWC, and an extended east region. The extended west region is mainly influenced by the South Asian Summer Monsoon (SASM) and the westerlies, while the extended east region is mainly influenced by the East Asian Summer Monsoon (EASM). The extended west, SWC, and extended east regions contribute 48.2%, 15.5%, and 24.5% of the moisture for the SWC precipitation, respectively. Moisture supply from the extended west region decreased at a rate of -
- 20 23.67.9 mmmon-1 decade-1 whereas that from the extended east increased at a rate of 4.21.4 mmmon-1 decade-1, resulting in an overall decrease of moisture supply. Further analysis reveals that the decline of summer-JAS precipitation is mainly caused by change in the seasonal-mean stationary component rather than the transient component of the moisture transport over the SWC region. In addition, tThe dynamic processes (i.e., changes in circulationwind) rather than the thermodynamic processes (i.e., changes in specific humidity) is-are dominant in affecting the seasonal-mean stationary-moisture transport. A
- 25 prevailing easterly anomaly of moisture transport that weakened moisture supply from the Indian Ocean is to a large extent responsible for the precipitation decrease over the SWC region.

**1** Introduction**

Frequent and severe droughts hit Southwest China (SWC) over the last decades with record-breaking events in the summer of 2006 and 2011, which had caused great losses to the society. The intensified drought is characterized by the persistent

deficit of precipitation (Wang et al., 2015b), and has attracted much attention (e.g. Barriopedro et al., 2012; Feng et al., 2014; Wang et al., 2015b; Tan et al., 2016; He et al., 2016; Zhang X. et al., 2017).

Many studies have analyzed the meteorological conditions that caused the extremely low precipitation for individual drought cases (e.g. Li et al., 2011; Lu et al., 2011; Yang et al., 2012). Taking the drought of summer 2006 as an example, a

- 5 stronger Western Pacific Subtropical High (WPSH) was found to lie anomalously northward and westward (Li et al., 2011). Under the direct control of WPSH, descent-descending flows-motion prevailed over SWC and the moisture transport from the Bay of Bengal (BOB) and South China Sea (SCS) was suppressed (Liu et al., 2009; Li et al., 2011). Further analysis revealed that the active convection over the Philippines and the weaker-than-normal heat source of the Tibetan Plateau drove the strengthened WPSH to shift northward and westward (Li et al., 2011). Meanwhile, a weak blocking high in the Ural
- 10 Mountains and a shallow East Asian trough facilitated a stronger-than-normal zonal circulation in the mid-latitudes, which hindered the intrusion of cold air into SWC (Zou and Gao, 2007). In summary, the configuration of the large-scale subtropical and mid-latitude circulations was unfavorable for the warm-moist air from the south and cold-dry air from the north to converge over SWC, and thus produced the severe drought.
- Some recent studies have endeavored to investigate the mechanisms causing the SWC drying in-from a long climatological perspective. Using stalagmite record as a proxy, Tan et al. (2016) found the period of 2009-2012 was the driest ever since 1760 AD in SWC. They further attributed the drying trend to the warming of Tropical Ocean, which had reduced the land-sea thermal gradient and the amount of moisture transported from the BOB. In another study, the possible influence of sea surface temperature (SST) in tropical Northwest Pacific (NWP) on the autumn precipitation in SWC was investigated (Wang et al., 2015a). It was found that the warm SST in NWP had likely contributed to the dry condition in SWC in recent decades.
  - Although previous studies have deepened our understanding of the SWC drying through attributing individual/general drought events or long-term precipitation trend to some probable causes, few of them have analyzed the changes in the precipitation moisture sources of this region. Tracing the precipitation-moisture sources not only can reveal the origin of moisture for precipitation (Gustafsson et al., 2010; Zhang C. et al., 2017; James et al., 2004; Sodemann and Zubler, 2010),
- 25 thus providing-it can also provide insights on to how the long term changes in moisture sources as well as how atmospheric circulations may affect precipitation in SWC. This study intends to identify changes in moisture sources of the SWC precipitation and study the relative changes in moisture transport during the last several decades and to investigate the possible mechanism of the SWC drying.

**2 Data, study area, and Methodology**

**30 2.1 Data and study area**

The reanalysis of the European Centre for Medium-Range Weather Forecasts (ECMWF) Interim (ERA-I hereafter) was used to calculate precipitable water and moisture flux (Dee et al., 2011; Trenberth, 1991). The ERA-I data adopted in this study

have a spatial resolution are of a grid of  $1.5^{\circ} \times 1.5^{\circ}$  grid cell, and the time span of the ERA I in this study is from 1979 to 2013. The data include the 6-h wind and moisture content at multiple levels from 200–1000 hPa, and surface pressure.

Due to the existing limitation with precipitation and evaporation estimates in reanalysis products (Trenberth et al., 2011; Tong et al., 2014), the ground-based 0.5°-gridded daily precipitation dataset from the China Meteorological Administration

5 (CMA) was used (Zhao et al., 2014; Zhao and Zhu, 2015). The CMA dataset is based on surface observations of ~2400 stations over China (Fig 1a). The gauge data are quality controlled but not homogenized. According to an estimation by Shen and Xiong (2015), the inhomogeneous stations takes up about 1.46% of all the stations.- The 3-h 1°-gridded evaporation fields from the Community Land Model in the Global Land Data Assimilation Systems (GLDAS, Rodell et al., 2004) dataset were usedchosen as GLDAS outperforms other reanalysis on surface variables (Wang and Zeng, 2012; Gao et al., 2014).

10 Over the ocean, the evaporation fields in ERA-I reanalysis were used directly, since there is no alternative estimate. The study area of SWC mainly encompasses three provinces of Sichuan, Yunnan, and Guizhou, and one municipality of Chongqing (Fig. 1a).It sits in the southeast foot of the Tibetan Plateau. In north SWC, the eastern Sichuan and Chongqing form the Sichuan Basin, while in south SWC, Yunnan and Guizhou form the Yun-Gui Plateau with an average altitude of around 2 km. The topographic height data are provided by the Global Land One-km Base Elevation Project (GLOBE).

**15 2.2 Water Accounting Model**

20

The Water Accounting Model (WAM) is an Eulerian model on moisture recycling, which can quantify the moisture sourcesink relations between evaporation and precipitation by tracking moisture forward or backward in time (van der Ent et al., 2010; van der Ent and Savenije, 2011; Keys et al., 2012). It is quite different from those Lagrangian models such as FLEXPART and HYSPLIT which track moisture based on the particle trajectories (Stohl and James, 2004; 2005; Sodemann et al., 2008; Draxler and Hess, 1998). In this study, moisture backtracking of WAM was applied to track the moisture origins of and their changes with the SWC precipitation. The algorithm is briefly described as follows.

[revised manuscript text omitted]

$$\overline{\mathbf{Q}} = \frac{1}{g} \int_{P_i}^{P_s} \overline{q} \overline{\mathbf{V}} dp + \frac{1}{g} \int_{P_i}^{P_s} \overline{q' \mathbf{V}'} dp, \qquad (1)$$
Stationary Transient

- 25 where Q is the vertically integrated moisture flux, g is the acceleration of gravity, q is the specific humidity, V is the horizontal wind vector,  $P_s$  is the surface pressure,  $P_t$  is the pressure at the top of the troposphere. The bars denote the monthly mean of variables, which is calculated using the average of 6-hourly values in each month. The apostrophes denote the anomalies of 6-hourly values to their monthly mean. The stationary component is the monthly mean moisture transported by the monthly mean flowwind, while the transient component is the transient moisture transported by the transient eddies. 30 The stationary component was calculated using the average of 6-h values in each month. The transient component was
- calculated using the 6-h deviations from the monthly mean.

The fluctuation of the stationary component in expression of divergence can be further expressed bydecomposed into a thermodynamic and a dynamic terms (Eq. (2); Seager et al., 2010; Li L. et al., 2013a).

$$\delta\left(\int_{P_{t}}^{P_{s}}\nabla\cdot\left(\overline{q}\overline{\mathbf{V}}\right)dp\right)\approx\underbrace{\int_{P_{t}}^{P_{s}}\nabla\cdot\left(\overline{q}_{a}\overline{\mathbf{V}}_{c}\right)dp}_{\text{Thermodynamic}}+\underbrace{\int_{P_{t}}^{P_{s}}\nabla\cdot\left(\overline{q}_{c}\overline{\mathbf{V}}_{a}\right)dp}_{\text{Dynamic}}, \tag{2}$$

where ∇· is the divergence operator and the divergence is calculated directly from the field of moisture flux; δ denotes the
fluctuation of the stationary component to its climatology; the subscript "c" denotes climatology and "a" the interannual deviation from the climatology; qc and Vc are the 35-year climatology of monthly mean specific humidity and wind velocity, respectively; qa and Va are the deviations from the 35-year climatology of each month. The thermodynamic (dynamic) component was-is solely determined by the changes in specific humidity (wind velocity) and thus represents the thermodynamic (dynamic) contribution (Li L et al., 2013a). The variations in stationary and transient components and thermodynamic terms were analyzed.

**3 Results and discussion**

Figure 1a-1b shows the annual precipitation trends from 1979–2013 calculated from the CMA gridded precipitation over the SWC region as marked out by the red box. The SWC precipitation has experiencedshows a declining trend in recent decades. The area-averaged annual SWC precipitation has decreased significantly with a rate of -2.72 mm yr-2+ (Fig. 1b2a). Table 1
provides the monthly precipitation trends during 1979–2013. It shows that the monthly trends from March to May are positive, although they are not statistically significant. The decreasing trends are the largest in the summer months of July, August, and September (JAS) with rates of -0.5, -1.1, and -1.0 mm month-1 yr-1 in July, August, and September, respectively. The decreasing trend is statistically significant at the 6% (1%) level in August (September). The total summer\_JAS precipitation decreased significantly at the 5% level with a rate of -2.570.86 mm mon-1 yr-1 (Fig. 1e2b). The precipitation

20 series with ERA-I over SWC are also shown in Fig. 2. It is evident that ERA-I estimates have large biases which are much larger than those with CMA, especially at annual scale. However, the two series both show similar decreasing patterns and comparable trend magnitudes with the CMA precipitations., which accounts for 94.5% of the annual trend. Since As the summer-JAS precipitation change-trend accounts for a major share of the annual precipitation trend (94.5% with CMA data), the analysis below will focus on the summer-JAS months.

**25 **3.1** Moisture origin and the trend in moisture contribution**

The climatological moisture contributions from the source grids in summer-JAS and their trends during 1979–2013 are shown in Fig. 23. The major moisture contributing region, i.e., grids with contribution over 0.8-27 mm mon-1-yr-1, are marked out (Fig. 2a3a), where 88.3% of summer-JAS precipitation moisture in SWC comes from. As shown in Fig.

2Generally, the farther away from the target region, the lower intensity of moisture is contributed to the target (Zhang C. et al., 2017; Keys et al., 2012; 2014). Yet, the lapse rate of moisture contribution intensity differs in different directions. The moisture contribution intensityIt decreases lapses slowly to the southwest and southeast, where moisture is transported by the Asian monsoons, indicating that the monsoon regions provide considerable amount of precipitation-moisture to SWC. This

- 5 result is consistent with Drumond et al. (2011), who traced precipitation moisture in Yunnan province from April to September and found two strong moisture sources of the Arabian Sea and the BOB, respectively. In contrast, the intensity lapses decreases rapidly to the north, suggesting that little precipitation moisture originates from the north. To the west of the SWC region, the intensity of moisture contribution is low in the dry lands area such as the Middle East but is relatively high in surrounding wet areas such as the Caspian Sea and the Red Sea, suggesting that humid regionthese water bodies tend to provides more moisture than arid the neighbouring dry regionlands.
  - As the moisture contribution trends show an opposing opposite pattern in the west and east (Fig. 2b3b), the major moisture contributing region is divided into three regions, namely the extended west, SWC, and the extended east regions. The extended west region covers an area west and southwest to SWC, and the extended east region covers an area east to SWC and a part of the Indian Ocean. Figure S1 shows the climatological moisture transport from July to September. It
- 15 indicates moisture from the extended west region largely enters the western and southern borders of SWC whereas moisture from the extended east region enters the eastern border via a route through the SCS. Moisture from the extended west region is likely affected by the South Asian Summer Monsoon (SASM) and the westerlies, while that from the extended east region is likely affected by the East Asian Summer Monsoon (EASM). When summed over regions, the extended west, SWC, and the extended east regions contribute 48.2%, 15.5%, and 24.5% of the total precipitation moisture, respectively. As SWC
- 20 situates eastward and downwind of the Tibetan Plateau (Fig. S1), moisture from regions to the west of the Plateau is mainly blocked by the Plateau, while the Plateau itself serves as a more important moisture source. According to statistics, The moisture contribution directly from the Tibetan Plateau\_icontributes only around 11.5% of the SWC precipitation, much-less than that from the SASM and EASM regions (Yao et al., 2012; Huang and Cui, 2015). Huang and Cui (2015) also notified the important role of Tibetan Plateau as a major source to provide moisture for precipitation in the Sichuan Basin. As the
- 25 Basin situates in north SWC, south SWC is, however, more accessible to the monsoons (Drumond et al., 2011). Thus, it is reasonable that the monsoons, which bring abundant moisture, contribute primary moisture to the JAS precipitation in SWC, while the westerlies contribute secondarily.

Figure 2b shows the trend of moisture contributed to the SWC summer precipitation from different regions during 1979 2013. Moisture supply (i.e., moisture contributed to the SWC precipitation) from most of the extended west region

[revised manuscript text omitted]

**25 4 Sensitivity analysis**

**4.1 On WAM**

30

In the study, the one vertical layer version of WAM (WAM1) was applied. WAM1 uses the vertically integrated fluxes with the moisture being well-mixed within the atmospheric column. As a matter of fact, "well-mixed" conditions of tagged atmospheric moisture are usually not met (Bosilovich, 2002; Goessling and Reick, 2013). At the same time if the horizontal winds are sheared vertically in direction, vertical inhomogeneities will generate, which may lead to substantial errors with 2-D moisture tracking models (Goessling and Reick, 2013). van der Ent et al. (2013) advanced WAM1 to the two vertical layer WAM (WAM2) that satisfyingly solved this problem and gave a simple metric to assess wind shear on when to use which model. The equations on the horizontal moisture flux shear following van der Ent et al. (2013) are

$$F_{z} = \frac{\left| \int_{P_{i}}^{P_{x}} qudp \right|}{\int_{P_{i}}^{P_{x}} |qu|dp}$$
(3)

and

5

$$F_m = \frac{\left| \int_{P_i}^{P_s} qv dp \right|}{\int_{P_i}^{P_s} |qv| dp}, \qquad (4)$$

where  $F_{\varepsilon}$  and  $F_m$  represent the zonal and meridional moisture flux shear, respectively. It can be easily judged that the flux shear value falls on a range between 0 and 1. The lower the value, the stronger the moisture flux shear. The climate means of horizontal moisture flux shear factors in JAS from 1979-2013 are shown in Fig. 8. The areal-weighted shear factor at the zonal direction over JAS SWC during 1979-2013 is 0.72, while it is 0.76 at the meridional direction. Taking the zonal shear

10 factor as an example, a shear factor of 0.72 means that 86% of the water goes in one direction with 14% in the opposite direction. Because the dominant moisture flux has a high share of the overall flux, the moisture flux shear is rather small in this case.

To further verify the applicability of WAM1, the year 1986 with the strongest moisture flux shear (the averaged zonal and meridional shear factor in JAS 1986 is 0.71) was selected to perform an inter-model comparison between WAM1 and

15 WAM2. As the atmospheric input data for WAM2 are model-level based, additional suite of ERA-I model-level atmospheric data in 1986 was prepared. The moisture contribution for JAS precipitation in 1986 SWC with WAM1 and WAM2 is shown in Fig. 9. It demonstrates that the spatial patterns of moisture contribution between WAM1 and WAM2 match quite well with each other.

**4.2 On ERA-I data**

- 20 ERA-I, as a modern reanalysis, has significantly improved in comparison to its prior version, ERA-40 (https://climatedataguide.ucar.edu/climate-data/era-interim; Trenberth et al., 2011). The ERA-I variables differ according to whether they are produced by the analysis or the forecast. The analysis fields are constrained by the observations while the forecast are produced by the model (Berrisford et al., 2011; Dee et al., 2011). Thus, observation-constrained fields as humidity, wind, etc., tend to be more reliable than those from model forecast as precipitation, evaporation, etc. (Berrisford et al., 2011).
- 25 al., 2011), so does the moisture transport derived from humidity and wind directly. In a comparison among several reanalyses (Trenberth et al., 2011), the long-term variation in moisture transport with ERA-I is rather stable which gives us more confidence in its application.

As in the study, observation-based precipitation (from CMA) and evaporation (GLDAS, forced with precipitation gauge observations) instead of their ERA-I forecast counterparts were used. On one hand, the input data for WAM becomes more accurate which facilitates more accurate results. On the other hand, changes in the ERA-I water cycle may induce changes in moisture origin and may further affect the trend results. In that consideration, moisture tracking for the SWC

- 5 precipitation with the original ERA-I evaporation and precipitation is also performed. The basic results are shown in Fig. 10. The basic patterns of moisture contribution with different E and P are similar (cf. Fig. 3a and 10a), except that sources with ERA-I E and P tend to contribute more moisture, since the SWC precipitation in JAS is higher with ERA-I than with CMA (see Fig. 2b). The major region (enclosed by the 0.27 mm mon-1 red line in Fig. 3a) contributes 89.4% with ERA-I E and P. The trend patterns are generally the same between both datasets (cf. Fig. 3b and Fig. 10b). Though there are small
- 10 differences in the magnitude of the rate and sometimes in rate signs over a few grids, the "East-increase and West-decrease" pattern remains unchanged. Thus, the major conclusions based on results with changed E and P remain unchanged. Instead, the application of CMA and GLDAS data tend to provide more reliable estimations. The small influence of evaporation and precipitation on the other hand highlight the importance of moisture transport. It is mainly due to the change in moisture transport that redistributes moisture, and leads to changes in moisture contributed to SWC as well as precipitation there.

**15 45 Conclusions**

Summer JAS precipitation over SWC has decreased significantly during 1979–2013. By tracing moisture\_the\_origins of moisture for JAS precipitation in the summer months (July, August, and September) and by analyzing the variations of moisture transport to SWC, we came to the following conclusions.

- Most moisture for the SWC summerJAS precipitation in SWC originates from in regions from the northern Indian
   Ocean to SWC and from South China Sea to SWC the Asian monsoon regions. The westerlies play a secondary role in supplying moisture. The extended west region, SWC, and the extended east region contributes 48.2%, 15.5% and 24.5% of moisture to the JAS precipitation in SWC-summer precipitation, respectively. The Tibetan Plateau region contributes a small portion (11.5%) of the precipitation-moisture for precipitation.
- (2) The decrease in the summer-JAS precipitation is mainly attributed to the reduced moisture supply from the extended west region. Moisture supply from the extended west region has decreased at a high rate (-23.67.9 mm mon-1 decade-1), and that from the extended east has increased at a low rate (4.21.4 mm mon-1 decade-1), resulting in an overall decrease of the moisture supply.

(3) The change in the stationary component has reduced moisture transport into SWCin JAS whereas the change in transient component has increased moisture transport in summer-during 1979–2013. The dynamic processes (i.e., changes in specific humidity) in affecting the precipitation. A prevailing easterly anomaly of moisture transport that weakened moisture transport supply from the Indian

Ocean is mainly responsible for the decrease of the SWC precipitation. The change in circulation may is likelybe related to the recent sea surface temperature change and need further investigation.

**56 Data availability**

The ERA-I data are available from supplied by the European Centre for Medium Range Weather Forecasts (ECMWF) and

5 are accessible at http://apps.ecmwf.int/datasets/data/interim-full-daily/. The GLDAS data are supplied byavailable from the NASA Goddard Earth Sciences Data and Information Services Center (GES DISC) and are accessible at https://disc.sci.gsfc.nasa.gov/services/grads-gds/gldas. The CMA precipitation dataset is provided by China Meteorological Data Service Center (CMDC) and is accessible at http://data.cma.cn/en. The GLOBE elevation data are downloaded from https://www.ngdc.noaa.gov/mgg/topo/globe.html.

10

**Competing interests*. The authors declare that they have no conflict of interest.**

Acknowledgements. This work was supported through the National Natural Science Foundation of China (41425002), the Key Research Program of the Chinese Academy of Sciences (ZDRW-ZS-2016-6-4), and the National Youth Top-notch Talent Support Program in China. Support from Swedish VR, STINT, BECC, MERGE and SNIC through S-CMIP are also 15 acknowledged.

20

 Table 1. The monthly precipitation trends (mm\_month-1 decade-1) in SWC during 1979 to 2013. The P-values of the trends were calculated based on the two-tailed Student's t-test.

| Month   | Jan  | Feb  | Mar  | Apr  | May  | Jun  | Jul  | Aug   | Sep   | Oct  | Nov  | Dec  |
|---------|------|------|------|------|------|------|------|-------|-------|------|------|------|
| Trend   | 0.2  | -1.4 | 1.3  | 0.1  | 3.7  | -1.9 | -5.0 | -10.7 | -10.0 | -1.6 | -1.4 | -0.5 |
| P-value | 0.79 | 0.27 | 0.41 | 0.97 | 0.28 | 0.50 | 0.32 | 0.06  | 0.01  | 0.55 | 0.49 | 0.61 |

15

---

## Author Response (AR2)

We truly appreciate the comments and suggestions. We have revised the manuscript accordingly. The point-by-point responses are given below.

Comments to the Author from the Co-Editor

Both reviewers agree that you have well addressed their comments and find your paper nearly ready for publication. However, reviewer #2 has a few suggestions for further small changes, which I think you should take into account for a final submission of your paper.

*A: We have made necessary changes to incorporate reviewer #2's suggestions in the revision.*

Reviewer #2

1. Reply to Comment 2: Section 2.2 should be updated with part of the response given in the reply

*A: We have updated Section 2.2 with the parts of the response given in the previous reply (P4, L15-22).*

2. Pg. 9 L. 10: a link to ucar is given here pointing to non-peer review material. Plese refer to peer-review work which is available on this subject.

*A: The link has been replaced by the peer-review reference (P9, L16).*

3. Reply to Comment 4: Please mention in Sec. 3.1 or elsewhere that the choice of the threshold contour is a subjective element of your study. The 50% contour is now only mentioned in the caption of Fig. 2.

*A: We have clarified the choice of the threshold contour in Section 3.1 (P5, L28-29). The 0.27 mm mon$^{-1}$ threshold contour is plotted to highlight the area contributing most (88.3%) of moisture to the study area. We plotted the 50% contour to show the core area contributing half of moisture to the study area. However, the core area is obvious in figure 2 so we decided to remove the 50% contour in the revision (P18, L4-5).*

**Tracing changes in atmospheric moisture supply to the drying Southwest China**

Chi Zhang[1], Qiuhong Tang[1,5], Deliang Chen[2], Laifang Li[3], Xingcai Liu[1], Huijuan Cui[4]

[1]Key Laboratory of Water Cycle and Related Land Surface Processes, Institute of Geographic Sciences and Natural Resources Research, Chinese Academy of Sciences, Beijing, China

[2]Regional Climate Group, Department of Earth Sciences, University of Gothenburg, Gothenburg, Sweden

[3]Earth and Ocean Sciences, Nicholas School of the Environment, Duke University, Durham, USA

[4]Key Laboratory of Land Surface Pattern and Simulation, Institute of Geographic Sciences and Natural Resources Research, Chinese Academy of Sciences, Beijing, China

[5]University of Chinese Academy of Sciences, Beijing, China

*Correspondence to*: Qiuhong Tang (tangqh@igsnrr.ac.cn)

**Abstract.** Precipitation over Southwest China (SWC) has significantly decreased during 1979–2013. The months from July to September (JAS) contributed the most to the decrease of precipitation. By tracing moisture sources of JAS precipitation over the SWC region, it is found that most moisture originates in regions from the northern Indian Ocean to SWC and from South China Sea to SWC. The major moisture contributing area is divided into an extended west region, SWC, and an extended east region. The extended west region is mainly influenced by the South Asian Summer Monsoon (SASM) and the westerlies, while the extended east region is mainly influenced by the East Asian Summer Monsoon (EASM). The extended west, SWC, and extended east regions contribute 48.2%, 15.5%, and 24.5% of the moisture for the SWC precipitation, respectively. Moisture supply from the extended west region decreased at a rate of -7.9 mm mon$^{-1}$ decade$^{-1}$ whereas that from the extended east increased at a rate of 1.4 mm mon$^{-1}$ decade$^{-1}$, resulting in an overall decrease of moisture supply. Further analysis reveals that the decline of JAS precipitation is mainly caused by change in the seasonal-mean component rather than the transient component of the moisture transport over the SWC region. In addition, the dynamic processes (i.e., changes in wind) rather than the thermodynamic processes (i.e., changes in specific humidity) are dominant in affecting the seasonal-mean moisture transport. A prevailing easterly anomaly of moisture transport that weakened moisture supply from the Indian Ocean is to a large extent responsible for the precipitation decrease over the SWC region.

**1 Introduction**

Frequent and severe droughts hit Southwest China (SWC) over the last decades with record-breaking events in the summer of 2006 and 2011, which had caused great losses to the society. The intensified drought is characterized by the persistent deficit of precipitation (Wang et al., 2015b), and has attracted much attention (e.g. Barriopedro et al., 2012; Feng et al., 2014; Wang et al., 2015b; Tan et al., 2016; He et al., 2016; Zhang X. et al., 2017).

Many studies have analyzed the meteorological conditions that caused the extremely low precipitation for individual drought cases (e.g. Li et al., 2011; Lu et al., 2011; Yang et al., 2012). Taking the drought of summer 2006 as an example, a stronger Western Pacific Subtropical High (WPSH) was found to lie anomalously northward and westward (Li et al., 2011). Under the direct control of WPSH, descending motion prevailed over SWC and the moisture transport from the Bay of Bengal (BOB) and South China Sea (SCS) was suppressed (Liu et al., 2009; Li et al., 2011). Further analysis revealed that the active convection over the Philippines and the weaker-than-normal heat source of the Tibetan Plateau drove the strengthened WPSH to shift northward and westward (Li et al., 2011). Meanwhile, a weak blocking high in the Ural Mountains and a shallow East Asian trough facilitated a stronger-than-normal zonal circulation in the mid-latitudes, which hindered the intrusion of cold air into SWC (Zou and Gao, 2007). In summary, the configuration of the large-scale subtropical and mid-latitude circulations was unfavorable for the warm-moist air from the south and cold-dry air from the north to converge over SWC, and thus produced the severe drought.

Some recent studies have endeavored to investigate the mechanisms causing the SWC drying from a long climatological perspective. Using stalagmite record as a proxy, Tan et al. (2016) found the period of 2009-2012 was the driest ever since 1760 AD in SWC. They further attributed the drying trend to the warming of Tropical Ocean, which had reduced the land-sea thermal gradient and the amount of moisture transported from the BOB. In another study, the possible influence of sea surface temperature (SST) in tropical Northwest Pacific (NWP) on the autumn precipitation in SWC was investigated (Wang et al., 2015a). It was found that the warm SST in NWP had likely contributed to the dry condition in SWC in recent decades.

Although previous studies have deepened our understanding of the SWC drying through attributing individual/general drought events or long-term precipitation trend to some probable causes, few of them have analyzed the changes in the precipitation moisture sources of this region. Tracing moisture sources not only can reveal the origin of moisture for precipitation (Gustafsson et al., 2010; Zhang C. et al., 2017; James et al., 2004; Sodemann and Zubler, 2010), it can also provide insight to long term change in moisture as well as how atmospheric circulations affect precipitation in SWC. This study intends to identify changes in moisture sources of the SWC precipitation and study the relative changes in moisture transport during the last several decades to investigate the possible mechanism of the SWC drying.

**2 Data, study area, and Methodology**

**2.1 Data and study area**

The reanalysis of the European Centre for Medium-Range Weather Forecasts (ECMWF) Interim (ERA-I hereafter) was used to calculate precipitable water and moisture flux (Dee et al., 2011). The ERA-I data adopted in this study are on a grid of 1.5° × 1.5°, and the time is from 1979 to 2013. The data include the 6-h wind and moisture content at multiple levels from 200–1000 hPa, and surface pressure.

Due to the existing limitation with precipitation estimates in reanalysis products (Trenberth et al., 2011; Tong et al., 2014), the ground-based 0.5°-gridded daily precipitation dataset from the China Meteorological Administration (CMA) was used (Zhao et al., 2014; Zhao and Zhu, 2015). The CMA dataset is based on surface observations of ~2400 stations over China (Fig 1a). The gauge data are quality controlled but not homogenized. According to an estimation by Shen and Xiong (2015), the inhomogeneous stations takes up about 1.46% of all the stations. The 3-h 1°-gridded evaporation fields from the Community Land Model in the Global Land Data Assimilation Systems (GLDAS, Rodell et al., 2004) dataset were chosen as GLDAS outperforms other reanalysis on surface variables (Wang and Zeng, 2012; Gao et al., 2014). Over the ocean, the evaporation fields in ERA-I reanalysis were used directly, since there is no alternative estimate.

The study area of SWC mainly encompasses three provinces of Sichuan, Yunnan, and Guizhou, and one municipality of Chongqing (Fig. 1a).It sits in the southeast foot of the Tibetan Plateau. In north SWC, the eastern Sichuan and Chongqing form the Sichuan Basin, while in south SWC, Yunnan and Guizhou form the Yun-Gui Plateau with an average altitude of around 2 km. The topographic height data are provided by the Global Land One-km Base Elevation Project (GLOBE).

**2.2 Water Accounting Model**

The Water Accounting Model (WAM) is an Eulerian model on moisture recycling, which can quantify the moisture source-sink relations between evaporation and precipitation by tracking moisture forward or backward in time (van der Ent et al., 2010; van der Ent and Savenije, 2011; Keys et al., 2012). It is quite different from those Lagrangian models such as FLEXPART and HYSPLIT which track moisture based on the particle trajectories (Stohl and James, 2004; 2005; Sodemann et al., 2008; Draxler and Hess, 1998). In this study, moisture backtracking of WAM was applied to track the moisture origins of and their changes with the SWC precipitation. The algorithm is briefly described as follows.

The input of WAM includes precipitation, evaporation, and atmospheric data (precipitable water and the vertically integrated moisture transport). The fallen precipitation in the target area was assumed to return to the air as "tagged water" in the model. The tagged water was mixed into the precipitable water with a ratio of $r$, which means only $r$ proportion of the precipitable water would finally fall into the target area. When it reverses the way back along the transport path, a certain amount of moisture, which is evaporated from the sources in the path, would fall into the target area. The ratio of that certain amount is also $r$. Taking the first source grid for example, it evaporates an amount of $e$ into the air at this time step. At the same time, the mixed ratio is $r$, and then the only $e*r$ would finally fall into the target area. The direct contribution from the grid at this time step is $e*r$. The tagged water would reduce the same amount of $e*r$ and move on to the next source grids till all the tagged water is depleted. By then, the total moisture contribution from each grid can be summed to produce a spatial distribution of moisture contributed to the precipitation in the target area.

As seen from the algorithm, the WAM is a 2-D model with the "well-mixed" assumption, where the tagged water mixes into the precipitable water sufficiently and the mixed ratio is independent of height. Though the well-mixed moisture conditions are not always met, a relatively low degree of vertical mixing suffices to maintain close to well-mixed conditions for the case of moisture flux with vertically uniform wind directions (Goessling and Reick, 2013). For the case of strong

directional shear of the horizontal moisture flux, a two-vertical-layer version of WAM was introduced by van der Ent et al. (2013) that solved the vertical inhomogeneities satisfyingly. The two-layer WAM is also implemented in the sensitivity analysis section as a validation to the one-layer WAM results.

The time step of WAM was set to 0.5 h for 1.5° grid in this study as in van der Ent et al. (2010) and van der Ent and Savenije (2011). The 6-h atmospheric data of precipitable water and moisture flux were linearly interpolated into the 0.5-h time step. For evaporation, the 1°-gridded GLDAS data were first interpolated into the 1.5° grid over the land, and then were merged with the ocean evaporation from the ERA-I reanalysis. The merged evaporation, which was 3-h accumulated, was divided equally into the 0.5-h time step. For precipitation, in order to reflect the diurnal cycle, the daily CMA and 3-h ERA-I precipitation fields were merged. The CMA precipitation was firstly transformed to the same spatial resolution as ERA-I by taking the means of the 0.5° grids that fell into the 1.5° grid. Then both monthly CMA and ERA-I precipitations were calculated for each 1.5° grid. By taking the monthly CMA value as norm, a rescaling factor $\varepsilon$ was produced for the monthly ERA-I value for each grid. All the ERA-I precipitation values (3 h) during a month within the grid were rescaled using the factor $\varepsilon$. Finally, the rescaled 3-h-accumulated ERA-I precipitation was equally distributed over the 0.5-h time step.

When  WAM  is applied at a monthly scale, a large amount of tagged water may be left in the air after one month's tracking rather than allocated to the surface sources. Many studies have shown that the average residence time of water vapor in the atmosphere is about 10 days (Trenberth, 1998; Numaguti, 1999; Trenberth, 1999). The ratio of tagged water to residence time is $e$-folding based. There would be $1/e$ (i.e., 36.8%) of the original water vapor left in the air after about 10 days. The backtracking in this study  is set to continue to run another 30 days with no input of precipitation. This setting would  make most (more than 95%) of the monthly precipitation moisture be allocated  to the surface sources. Taking July precipitation moisture tracking over the SWC region as an example, the tracked moisture accounts for 65.1% of precipitation on average from 1979-2013 if the tracked period is in July only, while it accounts for 97.4% if the tracked period covers June and July.

**2.3 Decomposing moisture transport**

To further understand the change of moisture transport in association with the change of moisture origin, the monthly vertically integrated moisture flux was decomposed into a stationary component and a transient component (Eq. (1); Li L. et al., 2013).

$$\overline{\mathbf{Q}} = \underbrace{\frac{1}{g}\int_{P_t}^{P_s}\overline{q}\overline{\mathbf{V}}dp}_{\text{Stationary}} + \underbrace{\frac{1}{g}\int_{P_t}^{P_s}\overline{q'\mathbf{V}'}dp}_{\text{Transient}}, \qquad (1)$$

where Q is the vertically integrated moisture flux, g is the acceleration of gravity, q is the specific humidity, V is the horizontal wind vector, $P_s$ is the surface pressure, $P_t$ is the pressure at the top of the troposphere. The bars denote the monthly mean of variables, which is calculated using the average of 6-hourly values in each month. The apostrophes denote

the anomalies of 6-hourly values to their monthly mean. The stationary component is the monthly mean moisture transported by the monthly mean wind, while the transient component is the transient moisture transported by the transient eddies.

The fluctuation of the stationary component in expression of divergence can be further decomposed into thermodynamic and dynamic terms (Eq. (2); Seager et al., 2010; Li L. et al., 2013).

$$\delta\left(\int_{P_t}^{P_s} \nabla \cdot \left(\overline{q\mathbf{V}}\right) dp\right) \approx \underbrace{\int_{P_t}^{P_s} \nabla \cdot \left(\overline{q}_a \overline{\mathbf{V}}_c\right) dp}_{\text{Thermodynamic}} + \underbrace{\int_{P_t}^{P_s} \nabla \cdot \left(\overline{q}_c \overline{\mathbf{V}}_a\right) dp}_{\text{Dynamic}}, \qquad (2)$$

where $\nabla \cdot$ is the divergence operator and the divergence is calculated directly from the field of moisture flux; $\delta$ denotes the fluctuation of the stationary component to its climatology; the subscript "c" denotes climatology and "a" the interannual deviation from the climatology; $\overline{q}_c$ and $\overline{\mathbf{V}}_c$ are the 35-year climatology of monthly mean specific humidity and wind velocity, respectively; $\overline{q}_a$ and $\overline{\mathbf{V}}_a$ are the deviations from the 35-year climatology of each month. The thermodynamic (dynamic) component is solely determined by the changes in specific humidity (wind velocity) and thus represents the thermodynamic (dynamic) contribution (Li L. et al., 2013).

**3 Results and discussion**

Figure 1b shows the annual precipitation trends from 1979–2013 calculated from the CMA gridded precipitation over the SWC region as marked out by the red box. The SWC precipitation shows a declining trend in recent decades. The area-averaged annual SWC precipitation has decreased significantly with a rate of -2.72 mm yr$^{-2}$ (Fig. 2a). Table 1 provides the monthly precipitation trends during 1979–2013. It shows that the monthly trends from March to May are positive, although they are not statistically significant. The decreasing trends are the largest in the summer months of July, August, and September (JAS) with rates of -0.5, -1.1, and -1.0 mm month$^{-1}$ yr$^{-1}$, respectively. The decreasing trend is statistically significant at the 6% (1%) level in August (September). The total JAS precipitation decreased significantly at the 5% level with a rate of -0.86 mm mon$^{-1}$ yr$^{-1}$ (Fig. 2b). The precipitation series with ERA-I over SWC are also shown in Fig. 2. It is evident that ERA-I estimates have large biases which are much larger than those with CMA especially at annual scale. However, the two series both show similar decreasing patterns and comparable trend magnitudes with the CMA precipitations. As the JAS precipitation trend accounts for a major share of annual precipitation trend (94.5% with CMA data), the analysis below will focus on the JAS months.

**3.1 Moisture origin and the trend in moisture contribution**

The climatological moisture contributions from the source grids in JAS and their trends during 1979–2013 are shown in Fig. 3. The major moisture contributing region, i.e., grids with contribution over 0.27 mm mon$^{-1}$, are marked out (Fig. 3a), where 88.3% of JAS precipitation moisture in SWC comes from. This threshold value is chosen subjectively to delineate the areas contributing most of JAS precipitation moisture to the SWC region. Generally, the farther away from the target region, the

lower intensity of moisture is contributed to the target (Zhang C. et al., 2017; Keys et al., 2012; 2014). Yet, the lapse rate of moisture contribution intensity differs in different directions. It decreases slowly to the southwest and southeast, where moisture is transported by the Asian monsoons, indicating that the monsoon regions provide considerable amount of moisture to SWC. This result is consistent with Drumond et al. (2011), who traced precipitation moisture in Yunnan province from April to September and found two strong moisture sources of the Arabian Sea and the BOB, respectively. In contrast, the intensity decreases rapidly to the north, suggesting that little moisture originates from the north. To the west of the SWC region, the intensity of moisture contribution is low in dry lands such as the Middle East but is relatively high in wet areas such as the Caspian Sea and the Red Sea, suggesting that these water bodies tend to provide more moisture than the neighbouring dry lands.

As the moisture contribution trends show an opposite pattern in the west and east (Fig. 3b), the major moisture contributing region is divided into three regions, namely the extended west, SWC, and the extended east regions. The extended west region covers an area west and southwest to SWC, and the extended east region covers an area east to SWC and a part of the Indian Ocean. Figure S1 shows the climatological moisture transport from July to September. It indicates moisture from the extended west region largely enters the western and southern borders of SWC whereas moisture from the extended east region enters the eastern border via a route through SCS. Moisture from the extended west region is likely affected by the South Asian Summer Monsoon (SASM) and the westerlies, while that from the extended east region is likely affected by the East Asian Summer Monsoon (EASM). When summed over regions, the extended west, SWC, and the extended east regions contribute 48.2%, 15.5%, and 24.5% of the total precipitation moisture, respectively. As SWC situates eastward and downwind of the Tibetan Plateau (Fig. S1), moisture from regions to the west of the Plateau is mainly blocked by the Plateau, while the Plateau itself serves as a more important moisture source. According to statistics, the Tibetan Plateau contributes around 11.5% of the SWC precipitation, less than that from the SASM and EASM regions. Huang and Cui (2015) also notified the important role of Tibetan Plateau as a major source to provide moisture for precipitation in the Sichuan Basin. As the Basin situates in north SWC, south SWC is, however, more accessible to the monsoons (Drumond et al., 2011). Thus, it is reasonable that the monsoons, which bring abundant moisture, contribute primary moisture to the JAS precipitation in SWC, while the westerlies contribute secondarily.

Moisture supply (i.e., moisture contributed to the SWC precipitation) from most of the extended west region experienced a decreasing trend of -7.9 mm mon$^{-1}$ decade$^{-1}$, accounting for 91.7% of the SWC precipitation trend, while that from most of the extended east region experienced an increasing trend of 1.4 mm mon$^{-1}$ decade$^{-1}$ (Fig. 3b). The trend of local moisture contribution to SWC precipitation is -0.4 mm mon$^{-1}$ decade$^{-1}$, accounting for 4.6% of the SWC precipitation trend, which suggests that change in local recycling played a minor role in the precipitation decrease over SWC.

[revised manuscript text omitted]

According to the calculated coefficients of determination, the dynamic component explains 80%, 81%, and 58% of the precipitation variances for July, August, and September, respectively. It confirms the dominant role of the dynamic processes in regulating the precipitation change in SWC. Indeed, the interannual variation in the SASM net precipitation (within the Arbian Sea-the Indian Subcontinent-BOB) is also dominated by the dynamic processes (Walker et al., 2015). It suggests that the dominant role played by the dynamic processes in regulating moisture transport and regional precipitation, not only validates in SWC, but prevails over a quite large area.

Figure 7 compares the dynamic component in JAS between the first and last ten years of the period of 1979–2013. There is an overall positive anomaly of moisture divergence over SWC with an easterly anomaly of moisture transport. Though there is a southwesterly anomaly of moisture transport from the Indian Ocean to the SWC direction in July and September, it does not contribute moisture transport to SWC because the anomaly ends on the south of the Tibetan Plateau. There is an easterly anomaly along the southern edge of the Tibetan Plateau, routing the moisture transport to the northern Indian subcontinent instead of the SWC region. The anomaly of moisture divergence, dynamically caused by the changes in circulation, is generally negative in the Indian subcontinent but positive in SWC (Tan et al., 2016). The prevailing easterly anomaly of moisture transport and pronounced regional anomalies of moisture divergence over SWC are likely to result from the change in the Asian summer monsoon system (Wei et al., 2014), which might be related to recent Pacific cooling and Indian Ocean warming (Ueda et al., 2015).

**4 Sensitivity analysis**

**4.1 On WAM**

In the study, the one vertical layer version of WAM (WAM1) was applied. WAM1 uses the vertically integrated fluxes with the moisture being well-mixed within the atmospheric column. As a matter of fact, "well-mixed" conditions of tagged atmospheric moisture are usually not met (Bosilovich, 2002; Goessling and Reick, 2013). At the same time if the horizontal winds are sheared vertically in direction, vertical inhomogeneities will generate, which may lead to substantial errors with 2-D moisture tracking models (Goessling and Reick, 2013). van der Ent et al. (2013) advanced WAM1 to the two vertical layer WAM (WAM2) that satisfyingly solved this problem and gave a simple metric to assess wind shear on when to use which model. The equations on the horizontal moisture flux shear following van der Ent et al. (2013) are

$$F_z = \frac{\left| \int_{P_t}^{P_s} qu\,dp \right|}{\int_{P_t}^{P_s} |qu|\,dp} \qquad (3)$$

and

$$F_m = \frac{\left| \int_{P_t}^{P_s} qv\,dp \right|}{\int_{P_t}^{P_s} |qv|\,dp}, \qquad (4)$$

where $F_z$ and $F_m$ represent the zonal and meridional moisture flux shear, respectively. It can be easily judged that the flux shear value falls on a range between 0 and 1. The lower the value, the stronger the moisture flux shear. The climate means of horizontal moisture flux shear factors in JAS from 1979-2013 are shown in Fig. 8. The areal-weighted shear factor at the zonal direction over JAS SWC during 1979-2013 is 0.72, while it is 0.76 at the meridional direction. Taking the zonal shear

5  factor as an example, a shear factor of 0.72 means that 86% of the water goes in one direction with 14% in the opposite direction. Because the dominant moisture flux has a high share of the overall flux, the moisture flux shear is rather small in this case.

To further verify the applicability of WAM1, the year 1986 with the strongest moisture flux shear (the averaged zonal and meridional shear factor in JAS 1986 is 0.71) was selected to perform an inter-model comparison between WAM1 and

10  WAM2. As the atmospheric input data for WAM2 are model-level based, additional suite of ERA-I model-level atmospheric data in 1986 was prepared. The moisture contribution for JAS precipitation in 1986 SWC with WAM1and WAM2 is shown in Fig. 9. It demonstrates that the spatial patterns of moisture contribution between WAM1 and WAM2 match quite well with each other.

**4.2 On ERA-I data**

15  ERA-I, as a modern reanalysis, has significantly improved in comparison to its prior version, ERA-40 (Dee et al., 2011; Trenberth et al., 2011). The ERA-I variables differ according to whether they are produced by the analysis or the forecast. The analysis fields are constrained by the observations while the forecast are produced by the model (Berrisford et al., 2011; Dee et al., 2011). Thus, observation-constrained fields as humidity, wind, etc., tend to be more reliable than those from model forecast as precipitation,

20  evaporation, etc. (Berrisford et al., 2011), so does the moisture transport derived from humidity and wind directly. In a comparison among several reanalyses (Trenberth et al., 2011), the long-term variation in moisture transport with ERA-I is rather stable which gives us more confidence in its application.

As in the study, observation-based precipitation (from CMA) and evaporation (GLDAS, forced with precipitation gauge observations) instead of their ERA-I forecast counterparts were used. On one hand, the input data for WAM becomes

25  more accurate which facilitates more accurate results. On the other hand, changes in the ERA-I water cycle may induce changes in moisture origin and may further affect the trend results. In that consideration, moisture tracking for the SWC precipitation with the original ERA-I evaporation and precipitation is also performed. The basic results are shown in Fig. 10. The basic patterns of moisture contribution with different E and P are similar (cf. Fig. 3a and 10a), except that sources with ERA-I E and P tend to contribute more moisture, since the SWC precipitation in JAS is higher with ERA-I than with CMA

30  (see Fig. 2b). The major region (enclosed by the 0.27 mm mon$^{-1}$ red line in Fig. 3a) contributes 89.4% with ERA-I E and P. The trend patterns are generally the same between both datasets (cf. Fig. 3b and Fig. 10b). Though there are small differences in the magnitude of the rate and sometimes in rate signs over a few grids, the "East-increase and West-decrease" pattern remains unchanged. Thus, the major conclusions based on results with changed E and P remain unchanged. Instead,

the application of CMA and GLDAS data tend to provide more reliable estimations. The small influence of evaporation and precipitation on the other hand highlight the importance of moisture transport. It is mainly due to the change in moisture transport that redistributes moisture, and leads to changes in moisture contributed to SWC as well as precipitation there.

**5 Conclusions**

JAS Precipitation over SWC has decreased significantly during 1979–2013. By tracing the origins of moisture for JAS precipitation and by analyzing the variations of moisture transport to SWC, we came to the following conclusions.

(1) Most moisture for the JAS precipitation in SWC originates in regions from the northern Indian Ocean to SWC and from South China Sea to SWC. The westerlies play a secondary role in supplying moisture. The extended west region, SWC, and the extended east region contributes 48.2%, 15.5%, and 24.5% of moisture to the JAS precipitation in SWC, respectively. The Tibetan Plateau region contributes 11.5% of the moisture for precipitation.

(2) The decrease in the JAS precipitation is mainly attributed to the reduced moisture supply from the extended west region. Moisture supply from the extended west region has decreased at a high rate ($-7.9$ mm mon$^{-1}$ decade$^{-1}$), and that from the extended east has increased at a low rate ($1.4$ mm mon$^{-1}$ decade$^{-1}$), resulting in an overall decrease of the moisture supply.

(3) The change in the stationary component has reduced moisture transport into SWC in JAS whereas the change in transient component has increased moisture transport during 1979–2013. The dynamic processes (i.e., changes in wind) are more important than the thermodynamic processes (i.e., changes in specific humidity) in affecting the precipitation. A prevailing easterly anomaly that weakened moisture transport from the Indian Ocean is mainly responsible for the decrease of the SWC precipitation. The change in circulation may be related to the recent sea surface temperature change and need further investigation.

**6 Data availability**

The ERA-I data are supplied by ECMWF and are accessible at http://apps.ecmwf.int/datasets/data/interim-full-daily/. The GLDAS data are supplied by the NASA Goddard Earth Sciences Data and Information Services Center (GES DISC) and are accessible at https://disc.sci.gsfc.nasa.gov/services/grads-gds/gldas. The CMA precipitation dataset is provided by China Meteorological Data Service Center (CMDC) and is accessible at http://data.cma.cn/en. The GLOBE elevation data are downloaded from https://www.ngdc.noaa.gov/mgg/topo/globe.html.

*Competing interests*. The authors declare that they have no conflict of interest.

*Acknowledgements*. This work was supported through the National Natural Science Foundation of China (41425002), the Key Research Program of the Chinese Academy of Sciences (ZDRW-ZS-2016-6-4), and the National Youth Top-notch

Talent Support Program in China. Support from Swedish VR, STINT, BECC, MERGE and SNIC through S-CMIP are also acknowledged.

**Table 1. The monthly precipitation trends (mm month$^{-1}$ decade$^{-1}$) in SWC during 1979 to 2013. The P-values of the trends were calculated based on the two-tailed Student's t-test.**

| Month | Jan | Feb | Mar | Apr | May | Jun | Jul | Aug | Sep | Oct | Nov | Dec |
|---------|------|------|------|------|------|------|------|-------|-------|------|------|------|
| Trend | 0.2 | -1.4 | 1.3 | 0.1 | 3.7 | -1.9 | -5.0 | -10.7 | -10.0 | -1.6 | -1.4 | -0.5 |
| P-value | 0.79 | 0.27 | 0.41 | 0.97 | 0.28 | 0.50 | 0.32 | 0.06 | 0.01 | 0.55 | 0.49 | 0.61 |

[Figure]

**Figure 1. (a) Geographic location of SWC (red box). The blue lines delineate the major provinces/municipality within SWC. (b) The trend of annual precipitation in SWC from 1979 to 2013 with the CMA precipitation. The circles indicate the 0.5° grids with at least one rain gauge in 1979. The red dots denote the precipitation trends with significance at the 5% level.**

[Figure]

**Figure 2. The time series of (a) annual and (b) July, August, and September (JAS) SWC precipitation from 1979 to 2013 with ERA-I (blue line) and CMA (black line) data. The trends are all significant at the 5% level based on the two-tailed Student's t-test.**

[Figure]

**Figure 3. (a) Climatology of the July, August, and September (JAS) moisture contribution to the SWC precipitation from 1979 to 2013. The red line delineates the major source region (i.e., grids with value above 0.27 mm mon⁻¹) which contributes 88.3% of the JAS precipitation moisture in SWC.**  **The black line divides the major source region into the extended west, SWC, and extended east regions. The blue line delineates the Tibetan Plateau (Zhang et al., 2014). (b) The trend of the JAS moisture contribution from 1979 to 2013. The dots indicate a trend at the 5% significance level based on the t test. Values outside the major region are not shown.**

[Figure]

[Figure]

**Figure 4. The difference of mean moisture contribution (unit: mm month⁻¹; shading) in July (a), August (b), and September (c) between 2004-2013 and 1979–1988. The vectors represent the difference of moisture transport.**

[Figure]

(a) Climatology of moisture divergence in SWC

[Figure]

(b) Moisture divergence series in SWC

**Figure 5. (a) Areal moisture divergence and its stationary and transient components over the SWC (unit: mm mon⁻¹) for July, August, and September during 1979–2013. (b) Moisture divergence in JAS and its stationary and transient components (unit: mm mon⁻¹) over the SWC during 1979–2013.**

[Figure]

**Figure 6. The anomalies of moisture divergence over JAS SWC (unit: mm mon⁻¹) caused by thermodynamic and dynamic terms during 1979–2013.**

[Figure]

**Figure 7. The monthly difference flux (vectors) of the dynamic component of the stationary moisture transport between the last and first 10 years (last – first) and its divergence (unit: $10^{-5}$ kg m$^{-2}$ s$^{-1}$; shading).**

[Figure]

**Figure 8. Horizontal moisture flux shear factor in JAS averaged over 1979–2013 with ERA-I. (a) Zonal moisture flux shear factor, (b) meridional moisture flux shear factor.**

[Figure]

**Figure 9. Moisture contribution of the SWC precipitation in JAS, 1986 with WAM1 (a) and WAM2 (b)**

[Figure]

**Figure 10. (a) Climatology of the JAS moisture contribution to the SWC precipitation from 1979 to 2013 with ERA-I E and P. (b) The trend of the JAS moisture contribution from 1979 to 2013 with ERA-I E and P. The red line and East-West division are the same as in Fig. 3.**